# SYP72 interacts with the mechanosensitive channel MSL8 to protect pollen from hypoosmotic shock during hydration

Xuemei Zhou [1,3], Yifan Zheng [1], Ling Wang [1], Haiming Li [1], Yingying Guo [1], Mengdi Li [1], Meng-xiang Sun [1] & Peng Zhao [1,2✉]

In flowering plants, hydration of desiccated pollen grains on stigma is a prerequisite for pollen germination, during which pollen increase markedly in volume through water uptake, requiring them to survive hypoosmotic shock to maintain cellular integrity. However, the mechanisms behind the adaptation of pollen to this hypoosmotic challenge are largely unknown. Here, we identify the Qc-SNARE protein SYP72, which is specifically expressed in male gametophytes, as a critical regulator of pollen survival upon hypoosmotic shock during hydration. SYP72 interacts with the MSCS-LIKE 8 (MSL8) and is required for its localization to the plasma membrane. Intraspecies and interspecies genetic complementation experiments reveal that *SYP72* paralogs and orthologs from green algae to angiosperms display conserved molecular functions and rescue the defects of Arabidopsis *syp72* mutant pollen facing hypoosmotic shock following hydration. Our findings demonstrate a critical role for SYP72 in pollen resistance to hypoosmotic shock through the MSL8 cascade during pollen hydration.

[1] State Key Laboratory of Hybrid Rice, College of Life Sciences, Wuhan University, 430072 Wuhan, China. [2] Hubei Hongshan Laboratory, 430070 Wuhan, China. [3] Present address: College of Life Sciences, South-Central University for Nationalities, 430074 Wuhan, China. ✉email: pzhao2000@whu.edu.cn

In flowering plants, each pollen grain contains two male gametes and has the potential to germinate to form a pollen tube that delivers the non-motile sperm cell to female gametophytes for fertilization[1]. Upon maturation, pollen grains of flowering plants are highly desiccated and metabolically quiescent[2]. When they land on the female stigma, desiccated pollen grains take up water and quickly become rehydrated. This initial phase is followed by metabolic activation of the pollen grains and rupture of the pollen coat through the combination of mechanical pressure and cell wall hydrolytic activities, leading to pollen germination and tube growth[3]. Hence, pollen hydration on the stigma is an essential prerequisite step for pollen germination. During this process, water uptake from the stigma by pollen grains dramatically increases their volume[4]. In Arabidopsis (Arabidopsis thaliana), for instance, the volume occupied by a pollen grain was shown to increase by 44% from 2 to 10 min of its landing on the stigma[5]. Water uptake and the resulting changes in volume create a hypoosmotic shock at the pollen grain plasma membrane. As a countermeasure to allow successful germination, over the course of evolution pollen grains must have established a responsive mechanism to enable the plasma membrane to remain flexible and maintain its integrity. However, the way that desiccated pollen grains adapt to this hypoosmotic shock during hydration to ensure pollen retention of cellular integrity and viability was largely unknown. A recent study reported that a plasma-membrane-localized mechanosensitive channel known as the mechanosensitive channel of small conductance (MSCS)-LIKE 8 (MSL8) is required for pollen grain survival of this hypoosmotic shock during rehydration[6]. However, it was not clear how MSL8 is regulated and whether other factors might be involved.

In both animals and plants, vesicle trafficking shuttles specific membrane proteins and cargo between the endomembrane system and the plasma membrane, in a process that is mediated by a large group of proteins known as SNAREs (soluble N-ethylmaleimide-sensitive factor attachment protein receptors)[7,8]. Animal and plant SNAREs contain an evolutionarily conserved SNARE motif comprising about 60–70 amino acids and are classified as Q-SNAREs and R-SNAREs based on the conserved glutamine (Q) or arginine (R) residue at the centre of the domain. Q-SNAREs may be further divided into three subfamilies, the Qa-, Qb-, and Qc-SNAREs[7,8]. In animals, SNARE proteins are key players in membrane repair and contribute to the specificity of membrane trafficking, cytokinesis, and mitochondrial division[8,9]. Similarly, plant SNARE proteins also play critical roles in several physiological processes, such as disease resistance and membrane fusion during cytokinesis[10,11]. Here we describe the role of the Qc-SNARE member SYNTAXIN OF PLANTS 72 (SYP72), whose encoding gene is specifically expressed in the male gametophyte, in pollen hydration. SYP72 directly binds to the mechanosensitive channel MSL8 to control its plasma membrane localization, which is critical for its channel activity. Strikingly, the molecular function of Qc-SNAREs in the SYP7 family is deeply conserved from green algae to angiosperms, as heterologous expression of various orthologues rescued the pollen viability defects seen in an Arabidopsis syp72 mutant in response to the hypoosmotic shock caused by rehydration. Our findings demonstrate a critical role for a Qc-SNARE in pollen survival of hypoosmotic shock that involves the formation of a SNARE–ion channel complex during pollen rehydration.

## Results

### SYP72 specifically accumulates in male gametophytes. Expression profiling analysis of Arabidopsis SNARE family members revealed that several SNARE genes are most highly expressed in mature pollen and are barely detected in vegetative organs (Supplementary Fig. 1). We focused on SYP72, a member of the plant-specific SYP7 gene family within the Qc-SNARE clade that is abundantly expressed in mature pollen and the pollen tube and had not previously been functionally characterized. Consistent with transcriptome deep sequencing (RNA-seq) data, SYP72 is specifically expressed in the male but not the female gametophyte, as evidenced by green fluorescence in pSYP72:H2B-GFP transgenic plants expressing a nuclear-targeted green fluorescent protein (GFP) driven by the SYP72 promoter, and pSYP72:GFP-SYP72 transgenic plants expressing a translational fusion of the GFP and SYP72 driven by the SYP72 promoter (Fig. 1a–d and Supplementary Fig. 2a, b). Indeed, we detected GFP fluorescence in pollen as early as the microspore stage, and it became exclusively localized to the vegetative cell at the tricellular pollen stage (Fig. 1a, b). By contrast, two other members of the SYP7 family, SYP71 and SYP73, showed very low transcript levels in mature pollen (Supplementary Fig. 1), an observation that we validated with GFP-SYP71 and GFP-SYP73 fusion constructs under the control of their respective endogenous promoters (Supplementary Fig. 2c–f). These results demonstrated that SYP72 is specifically expressed in the male gametophyte.

### SYP72 is required for pollen germination and male fertility. To explore the role of the SYP72 protein, we obtained three T-DNA insertion lines with insertions in SYP72 from the Arabidopsis Biological Resource Center (ABRC). We confirmed the positions of the T-DNA insertions by sequencing and the transcript levels of SYP72 in these mutants by reverse transcription–PCR (RT-PCR) (Supplementary Fig. 3a, b). The syp72 mutants grew normally, with no visible vegetative growth defects compared to Col-0 wild-type (WT) plants (Supplementary Fig. 4a). Reciprocal crosses between WT and heterozygous syp72 mutants showed that the transmission of mutant alleles through the pollen was significantly impaired, but transmission through the female gametophyte was not affected, suggesting that pollen function is severely compromised in syp72 mutants (Supplementary Table 1). To investigate whether the reduced male transmission rates seen in syp72 were caused by a pollen developmental defect, we identified plants homozygous for each T-DNA insertion and carefully inspected the morphology and viability of their mature pollen. Pollen grains from syp72 mutants were fully viable, as revealed by staining with Alexander's stain (Supplementary Fig. 4b–d). In addition, syp72 pollen grains were morphologically indistinguishable from WT pollen grains, with two generative nuclei and one vegetative nucleus (Supplementary Fig. 4e, f). These data indicated that syp72 pollen grains develop normally.

We next assessed the germination potential of syp72 pollen to test whether it might contribute to the reduced male transmission rates of syp72 mutants. Notably, loss of SYP72 function led to a marked decrease in pollen germination, with only about 1.3% of syp72-1 pollen grains initiating germination in vitro (Fig. 1e, f). We validated this observation in vivo by aniline blue staining, which stains callose within pollen tubes (Supplementary Fig. 4g). In addition, pollen tube growth of syp72 mutants was slower than that of WT plants on both the medium and the WT pistils (Fig. 1g and Supplementary Fig. 4h), but syp72 pollen tubes did not display visible defects in pollen tube guidance (Supplementary Fig. 4i). Consistent with their much lower pollen germination rate, we observed a significant reduction in seed set (29.6% seed setting) in plants homozygous for the syp72-1 allele, with 15.1 ± 1.2 seeds per silique, compared to 56.6 ± 2.4 seeds per silique in the WT (Fig. 1h, i). The seed setting defect of homozygous syp72-1 mutant plants was fully rescued both by a genomic fragment of SYP72 (gSYP72) and by the pSYP72:GFP-

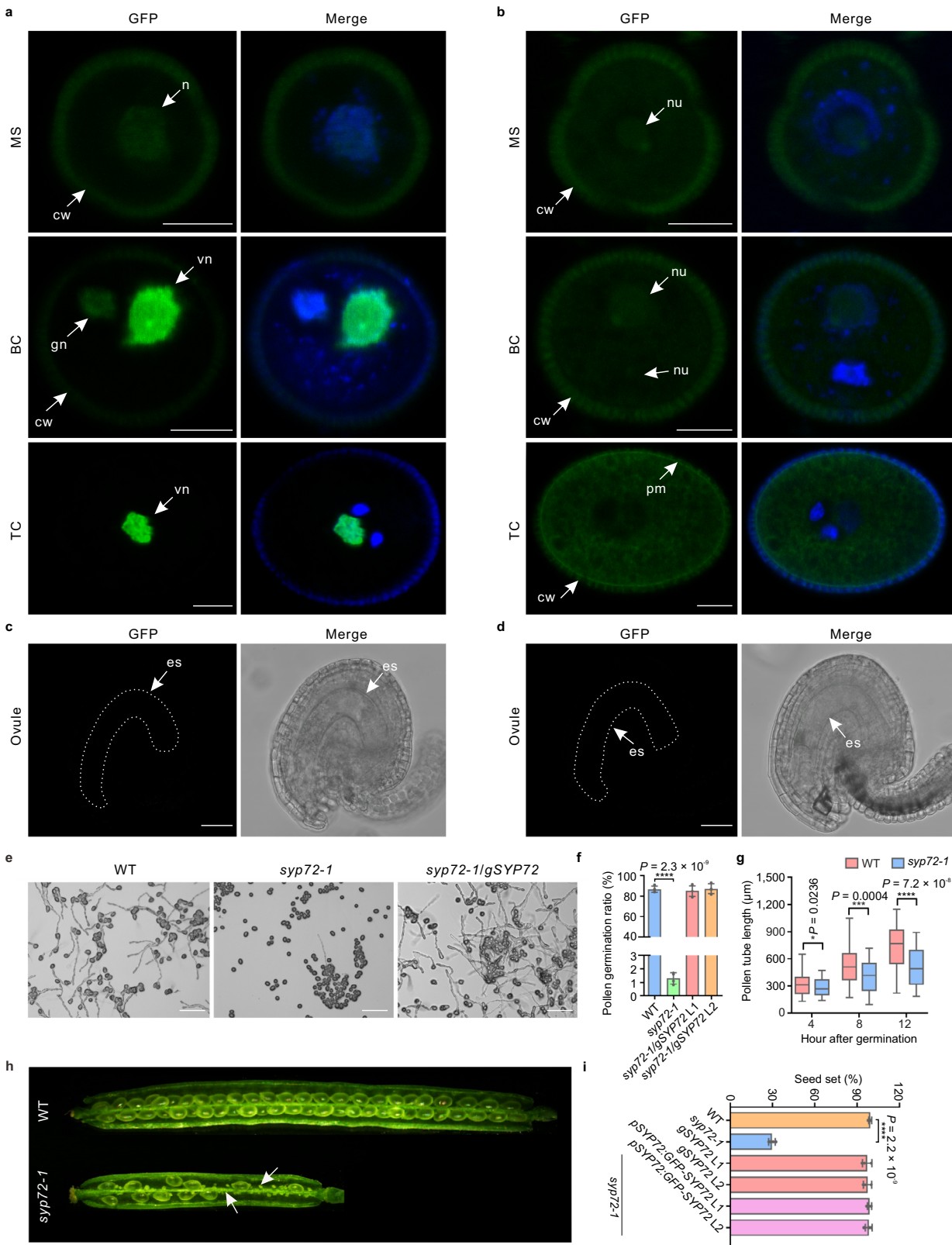

*SYP72* used above (Fig. 1i), demonstrating that (i) loss of SYP72 was responsible for the pollen phenotype and (ii) the GFP-SYP72 fusion protein was functional.

**SYP72 protects pollen from hypoosmotic shock during hydration.** Early reports demonstrated that pollen hydration on the stigma is an essential prerequisite to pollen germination[1].

Because the morphology and viability of *syp72* pollen grains were similar to those of WT plants, we next investigated whether *syp72* pollen was defective in its ability to survive the rehydration-induced hypoosmotic shock. We first examined the membrane integrity of desiccated pollen by double staining with fluorescein diacetate (FDA) and propidium iodide (PI) during the pollen rehydration process. FDA stains pollen with the integrated

**Fig. 1 SYP72 is specifically expressed in the vegetative cell of mature pollen, and syp72 mutants display pollen germination defects. a–d** Analysis of *pSYP72:H2B-GFP* (**a**, **c**) and *pSYP72:GFP-SYP72* (**b**, **d**) reporter lines reveals that *SYP72* is specifically expressed in the male (**a**, **b**) but not the female gametophyte (**c**, **d**). GFP GFP fluorescence, Merge merged images for GFP and 4′,6-diamidino-2-phenylindole (DAPI) staining in **a**, **b**, and merged images for GFP and differential interference contrast (DIC) in **c**, **d**. Images of non-transgenic controls are shown in Supplementary Fig. 2. Nucleolus of microspore and bicellular pollen have weak auto-fluorescence, and auto-fluorescence was also observed in the pollen wall. MS microspore, BC bicellular pollen, TC tricellular pollen, n nucleus, nu nucleolus, gn generative nucleus, vn vegetative nucleus, sn sperm cell nucleus, pm plasma membrane, es embryo sac. Scale bars: 5 μm for pollen, 25 μm for ovule. **e**, **f** Mutation in *SYP72* leads to a marked decrease in pollen germination. **e** Representative images of pollen germination from the Col-0 wild type (WT), the *syp72-1* mutant, and the *gSYP72* complementation line. Scale bars, 100 μm. **f** Pollen germination frequency in the WT, *syp72-1*, and the *gSYP72* complementation line. Data represent the mean ± SD from four independent experiments, with 300 pollen grains each ($n = 1200$). **g** Statistical data of pollen tube length of the WT and the *syp72-1* mutant at different hours after germination. Data for pollen tube length are presented in the box-and-whisker plots. The centreline in the plot represents the 50th percentile. The bottom and top of each box indicate 25th and 75th percentiles, and the whiskers represent the minimum and maximum values. (WT, $n = 80$ biologically independent samples; *syp72-1*, $n = 70$ biologically independent samples). **h** Representative images of siliques from the WT and the *syp72-1* mutant. Arrows indicate aborted seeds. **i** Quantification of seed setting from the WT, the *syp72-1* mutant, *gSYP72*, and *GFP-SYP72* complementation lines. Data represent the mean ± SD from four independent quantifications, with five siliques each ($n = 20$). Asterisks indicate statistically significant differences (two-tailed Student's *t* test, *$P < 0.05$; ***$P < 0.001$; ****$P < 0.0001$). Observation of GFP in *pSYP72:H2B-GFP* (**a**, **c**) and *pSYP72:GFP-SYP72* (**b**, **d**) transgenic plants was repeated at least three times with similar results.

plasma membrane, while PI stains only pollen with compromised plasma membrane integrity. We discovered that, in contrast to WT pollen, only a small proportion of *syp72* pollen grains were FDA positive and PI negative (Fig. 2a, b). Differential interference contrast (DIC) observation and plasma membrane integrity assay in the WT stigma further confirmed the rupture phenotype of *syp72* pollen during rehydration (Fig. 2c–f). Furthermore, the percentage of FDA-positive pollen grains in the WT remained constant over a 2-h incubation in distilled water, whereas the percentage of FDA-positive pollen grains in *syp72-1* decreased significantly from 15.3% at time 0 to 3.4% after 2 h (Fig. 2g), suggesting that SYP72 is required for pollen to survive the imposed hypoosmotic challenge and maintain cellular integrity in the process of rehydration.

As rehydration in distilled water is a relatively extreme condition for pollen grains, we then assessed the cellular integrity of pollen grains treated with different concentrations of polyethylene glycol (PEG) 3350. The fraction of FDA-positive *syp72* pollen grains increased with the percentage of PEG 3350 applied (Fig. 2h, i). In addition, the majority of *syp72-1* pollen grains dissected from anthers before desiccation survived the hypoosmotic challenge of rehydration (Fig. 2j, k). This latter phenotype of *syp72* pollen grains was rather similar to that seen in pollen lacking the mechanosensitive ion channel MSL8[6]. These results indicate that *syp72* pollen develops normally before dehydration, implying that the defect in pollen germination is attributable to an inability of the pollen to survive the hypoosmotic shock encountered during rehydration.

**SYP72 localizes to endosomes and plasma membrane.** To determine the intracellular localization of SYP72, we used the transgenic line *pSYP72:GFP-SYP72* described above for localization analysis. We visualized GFP-tagged SYP72 both at the plasma membrane and in intracellular compartments within pollen grains (Fig. 3 and Supplementary Fig. 5). To more precisely identify the cellular compartment to which SYP72 localized, we crossed a representative *SYP72:GFP-SYP72* transgenic line with a set of marker lines expressing *mCherry*- or *red fluorescent protein* (*RFP*)-tagged markers targeted to different organelles: the endosome markers Rab A1e, Rab A1g, Rab F2b, Rab A5d, Rab C1, and Rab D1; the trans-Golgi network/early endosome (TGN/EE) marker VESICLE TRANSPORT V-SNARE 12 (VTI12); the vacuolar marker VESICLE-ASSOCIATED MEMBRANE PROTEIN 711 (VAMP711); the plasma membrane marker NOVEL PLANT SNARE 12 (NPSN12)[12]; and the Golgi marker ST-RFP[13]. GFP-SYP72 clearly localized separately from the Golgi marker

ST-RFP and the vacuolar marker VAMP711, but it partially overlapped with the TGN/EE and endosome markers. GFP-SYP72 also showed a strong colocalization with NPSN12 at the plasma membrane, as expected (Fig. 3 and Supplementary Fig. 5). Correlation analysis of GFP-SYP72 with these organellar markers using Pearson's correlation coefficients confirmed this result. Indeed, we calculated a strong correlation between GFP-SYP72 and the plasma membrane marker NPSN12 ($r = 0.76 ± 0.06$) and the endosome marker Rab A1e ($r = 0.66 ± 0.12$). These results indicated that SYP72 localizes to the plasma membrane and post-Golgi endosomes.

**SYP72 directly binds the mechanosensitive channel MSL8.** The *syp72* mutant pollen exhibited a phenotype in distilled water similar to that of *msl8* mutants, which are defective in a plasma-membrane-localized mechanosensitive channel that is required for the maintenance of pollen cellular integrity upon hypoosmotic shock during hydration[6]. We therefore set out to investigate the possible biochemical and genetic connections between SYP72 and MSL8. We obtained three independent lines of evidence supporting the possibility that SYP72 can directly bind MSL8. We first used split-ubiquitin yeast two-hybrid assays in which MSL8 fused to the C-terminal half of yeast ubiquitin (Cub) served as bait and SYP72 fused to the N-terminal half of yeast ubiquitin NubG as prey. MSL9 and another Qc-SNARE SYP52 were used as the controls in the yeast two-hybrid assay. As positive and negative controls, MSL8-Cub was transformed with NubI or NubG, respectively. MSL8-Cub interacted with NubG-SYP72 and the positive control NubI, but not with the negative control NubG and the Qc-SNARE SYP52 (Fig. 4a and Supplementary Fig. 6a). To validate the interaction between SYP72 and MSL8, we performed co-immunoprecipitation (Co-IP) and pull-down assays. Immunoprecipitation of GFP-SYP72 with GFP-trap in both Arabidopsis pollen grains and tobacco leaves yielded a co-immunoprecipitating band corresponding to Myc-tagged MSL8. By contrast, immunoprecipitation of GFP in plants expressing the fluorescent protein alone did not yield a similar band detected by the anti-Myc antibody (Fig. 4b and Supplementary Fig. 6b). The pull-down assays also support the interaction of SYP72 with MSL8 (Fig. 4c). In addition, SYP72-GFP strongly colocalized with MSL8-RFP at both the plasma membrane and intracellular compartments of mature pollen (Fig. 4d, e). These results thus confirmed the binding ability of SYP72 to the mechanosensitive channel MSL8.

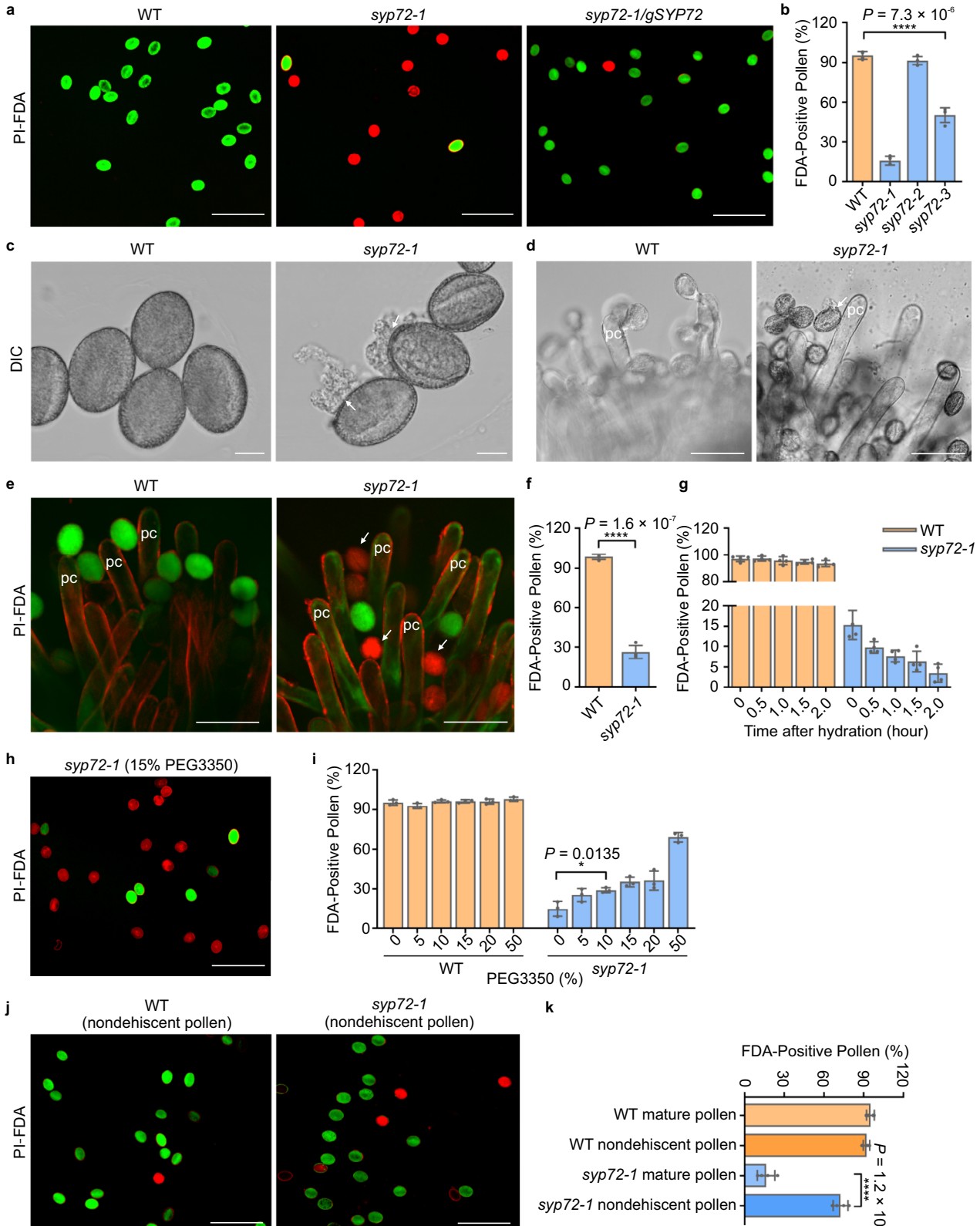

**SYP72 is required for MSL8 localization to the plasma membrane.** Given that SYP72 can directly bind to the mechanosensitive channel MSL8, we next tested whether the function of SYP72 in regulating pollen survival after hypoosmotic shock due to rehydration involves MSL8. First, we carefully quantified *MSL8* mRNA and MSL8 protein levels in the *syp72-1* mutant. Both were comparable between pollen from WT and *syp72-1* mutant plants

(Fig. 4f–h). As the localization of membrane ion channels to the plasma membrane is critical for their channel function, we then compared the intracellular location of MSL8 in *syp72-1* and WT. The intracellular location of MSL8 in *syp72* pollen was distinct from that in WT pollen: MSL8 localized to the plasma membrane in WT pollen, but not in the *syp72* mutant, where it instead exhibited a uniform intracellular distribution (Fig. 4i–k). This

**Fig. 2 SYP72 maintains cellular integrity during pollen rehydration. a, b** *syp72* mutation results in loss of plasma membrane integrity after rehydration. **a** Propidium iodide (PI) and fluorescein diacetate (FDA) double staining to assess plasma membrane integrity of mature pollen from the WT, the *syp72-1* mutant, and the *gSYP72* complementation line. Scale bars, 100 μm. **b** Frequency of FDA-positive pollen from the WT and three *syp72* T-DNA insertion lines. Data represent the mean ± SD from four independent assays, with 500 pollen grains scored each time ($n = 2,000$). **c** Representative DIC images showing ruptured *syp72* pollen during hydration ($n = 60$ biologically independent pollen grains). Scale bars, 10 μm. **d, e** DIC observation (**d**) and PI-FDA staining (**e**) assess plasma membrane integrity of mature pollen in the WT pistil during hydration. Scale bars, 50 μm. DIC observation of ruptured *syp72* pollen grains in the pistil was performed three times, with three pistils analysed in each experiment ($n = 9$). **f** Frequency of FDA-positive pollen from the WT and the *syp72-1* mutant in the WT pistil during hydration. Data represent the mean ± SD from four independent assays. ($n = 658$ pollen grains for WT and 528 for *syp72-1*). **g** Frequency of FDA-positive pollen from the WT and the *syp72-1* mutant during g rehydration time course in distilled water. Data represent the mean ± SD from four independent assays, with 400 pollen grains scored each time. **h, i** Polyethylene glycol (PEG) treatment increases the frequency of FDA-positive pollen from the *syp72-1* mutant during rehydration. **h** Representative images of PI-FDA double-stained *syp72-1* pollen after PEG treatment. Scale bars, 100 μm. **i** Frequency of FDA-positive pollen from the *syp72-1* mutant after PEG treatment. Data represent the mean ± SD from three independent assays, with 400 pollen grains scored each time. **j, k** PI-FDA double staining assessment of plasma membrane integrity in nondehiscent tricellular pollen grains dissected from anthers. **j** Representative images of PI-FDA-stained nondehiscent tricellular pollen. Scale bars, 100 μm. **k** Frequency of FDA-positive nondehiscent tricellular pollen grains from WT and *syp72-1* plants. Data represent the mean ± SD from four independent assays, with 400 pollen grains scored each time. pc papilla cell. Arrows indicate ruptured pollen. Two-tailed Student's *t* test was used for statistical analysis (*$P < 0.05$; ****$P < 0.0001$).

observation indicated that SYP72 is involved in trafficking MSL8 to the plasma membrane to exert its ion channel function during rehydration.

We next investigated which trafficking pathway responsible for MSL8 reaching the plasma membrane might be impaired in *syp72* pollen grains. We performed high-pressure freezing and freeze-substitution-based transmission electron microscopy (TEM) analysis to look for ultrastructural abnormalities in the *syp72* pollen grains. This analysis revealed no visible differences in Golgi morphology between *syp72* and WT pollen (Fig. 5a). By contrast, numerous vesicle-like structures accumulated in *syp72* but not the WT pollen, suggesting a defect in post-Golgi trafficking (Fig. 5a, b). To confirm the trafficking pathway impaired in *syp72* pollen, we introduced the *syp72* mutation into a series of organelle markers—SYP32 and ST-RFP (Golgi), VTI12 (TGN/EE), Rab D1 (post-Golgi), Rab A1g (a recycling marker), and Rab F2a and Rab F2b (late endosome)—to track organellar distribution in *syp72* pollen grains (Fig. 5c and Supplementary Fig. 7). Consistent with the TEM results, the morphology and distribution of Golgi showed no visible differences relative to the WT, whereas VTI12-labelled TGN/EE was aggregated, suggesting a defect in protein trafficking (Fig. 5c and Supplementary Fig. 7). These results revealed that SYP72 resides along a pathway from the Golgi to the plasma membrane and directly binds to MSL8 to facilitate its localization to the plasma membrane, which is essential for its membrane ion channel function.

**SYP7 SNAREs have conserved function for MSL8 localization.** Arabidopsis syntaxins can be grouped into eight clusters, known as SYP1–SYP8, based on their sequence similarity (Fig. 6a). Of those, the SYP7 family contains three members, SYP71, SYP72, and SYP73, and is unique to plants, with no orthologues in other kingdoms[14]. Protein sequence alignment confirmed that the three SNAREs in the SYP7 subfamily share high sequence similarity (Supplementary Fig. 8a). However, their encoding genes also displayed distinct expression patterns in vegetative and reproductive processes during plant development. Both RNA-seq data and reporter analysis with *GFP-SYP* constructs driven by their respective promoters revealed that only *SYP72* is expressed in mature pollen, whereas *SYP71* and *SYP73* expression was barely detectable in mature pollen (Fig. 1a, b and Supplementary Fig. 2), suggesting that each member of the SYP7 subfamily plays tissue-specific roles in plant development, with little redundancy. These interesting characteristics of SYP7 subfamily SNAREs prompted us to investigate whether the three SYP7 members might have a conserved molecular function in promoting MSL8 localization to the plasma membrane, even though they have different

expression patterns. To this end, we generated a series of constructs in which *GFP-SYP71* or *GFP-SYP73* constructs were placed under the control of the *SYP72* promoter (*pSYP72:GFP-SYP71* and *pSYP72:GFP-SYP73*) and introduced these into the *syp72-1* background to drive *SYP71* and *SYP73* expression in pollen. Like *SYP72*, both *SYP71* and *SYP73* localized to the plasma membrane of mature pollen (Fig. 6b) and successfully rescued the pollen burst phenotype of the *syp72-1* mutant in the process of rehydration, as measured by PI staining (Fig. 6c, d). In agreement with the complementation of the *syp72-1* mutant by the *pSYP72:GFP-SYP71* and *pSYP72:GFP-SYP73* transgenes, in vivo pollen germination and seed setting in the *syp72-1* mutant were also restored to WT levels by the ectopic expression of *SYP71* or *SYP73* in *syp72-1* pollen (Supplementary Fig. 9a–c). However, the expression of genes encoding SNARE proteins from the adjacent subgroup, *SYP5* and *SYP6*, did not rescue the phenotype of the *syp72-1* mutant (Fig. 6b–d and Supplementary Fig. 9a–c). Indeed, similar to the *syp72-1* mutant, *syp72-1* plants expressing *pSYP72:GFP-SYP51* or *pSYP72:GFP-SYP61* were characterized by a comparable pollen burst rate during rehydration (Fig. 6b–d).

Next, to determine whether SYP71 and SYP73 assist MSL8 in its localization to the plasma membrane for its channel activity in the *syp72-1* mutant, we introduced the *gMSL8-RFP* reporter into the *syp72-1 pSYP72:GFP-SYP71* and *syp72-1 pSYP72:GFP-SYP73* complementation lines. Consistent with the phenotypic results, we discovered that whereas MSL8 relocalized to the plasma membrane in the *syp72-1* mutant lines complemented by *SYP71* (*pSYP72:GFP-SYP71*) or *SYP73* (*pSYP72:GFP-SYP73*), it did not do so in *syp72-1* lines expressing *SYP51* or *SYP61* (*p72-1 pSYP72:GFP-SYP51* or *syp72-1 pSYP72:GFP-SYP61*) (Fig. 6e–g), further confirming that all three SNAREs in the SYP7 subfamily share the conserved molecular function of facilitating MSL8 localization to the plasma membrane.

**Conserved role of SYP7 in pollen hydration during evolution.** Because Arabidopsis SYP7 subfamily SNARE members play a conserved role in hypoosmotic shock survival during pollen rehydration, we explored the evolutionary origin of the SYP7 subfamily over the course of plant diversification. We surveyed SYP7 subfamily SNARE members in 15 plant species representing the alga, liverwort, moss, lycophyte, gymnosperm, and angiosperm lineages using Arabidopsis SYP72 as a protein query against publicly available sequences deposited at the National Center for Biotechnological Information (Fig. 7a and Supplementary Fig. 8). Strikingly, the counterparts of SYP72 appeared specific to the Brassicaceae, as they were present in

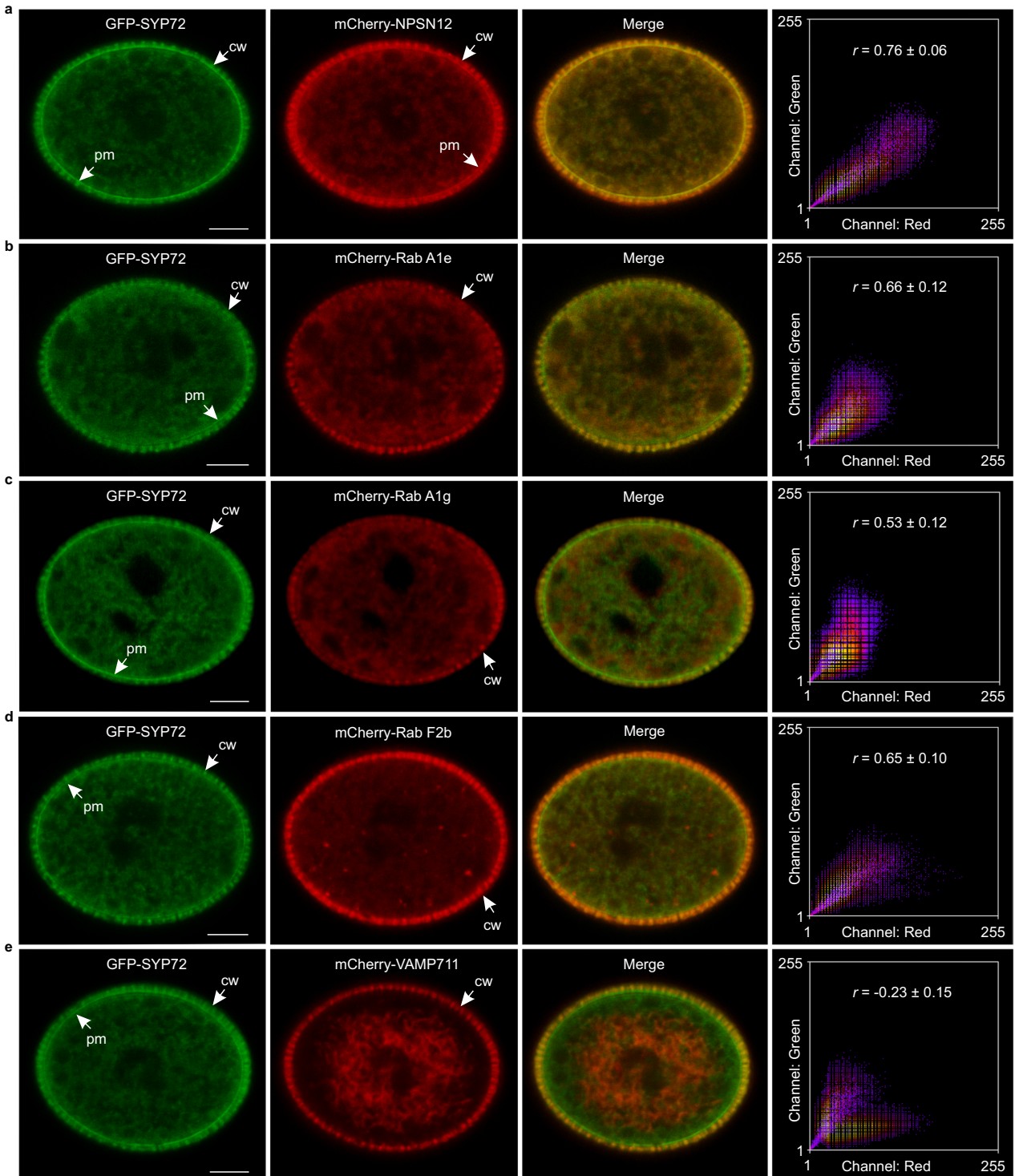

**Fig. 3 Intracellular localization of SYP72 to the plasma membrane and post-Golgi endosomes in Arabidopsis pollen. a–e** Representative confocal microscopic images showing the intracellular localization of GFP-SYP72 and various organellar markers in pollen grains: mCherry-NPSN12 (plasma membrane, **a**), mCherry-Rab A1e (recycling endosome, **b**), mCherry-Rab A1g (recycling endosome, **c**), mCherry-Rab F2b (late endosome, **d**), and mCherry-VAMP711 (vacuole, **e**). Scale bars, 5 μm. The rightmost column shows the scatterplots of the fluorescence intensity of the merged images. Pearson correlation coefficients (*r*) indicate the extent of colocalization between GFP-SYP72 and each organellar marker. Data represent mean ± SD (*n* = 15 biologically independent pollen grains for mCherry-NPSN12, mCherry-Rab A1e, mCherry-Rab A1g, and mCherry-VAMP711; *n* = 12 independent pollen grains for mCherry-Rab F2b). cw cell wall, pm plasma membrane.

lyrate rockcress (*Arabidopsis lyrata*), pink shepherd's purse (*Capsella rubella*), and rapeseed (*Brassica napus*), but not in other plants. However, the majority of the green lineage, including green algae, the moss *Physcomitrium* (*Physcomitrella*) *patens*, and

the liverwort *Marchantia polymorpha*, had orthologues of SYP71, another member of the SYP7 subfamily (Supplementary Fig. 8a, b). According to the phylogenetic tree, these SYP7 subfamily proteins clustered into six clades. In angiosperms, SYP72 and

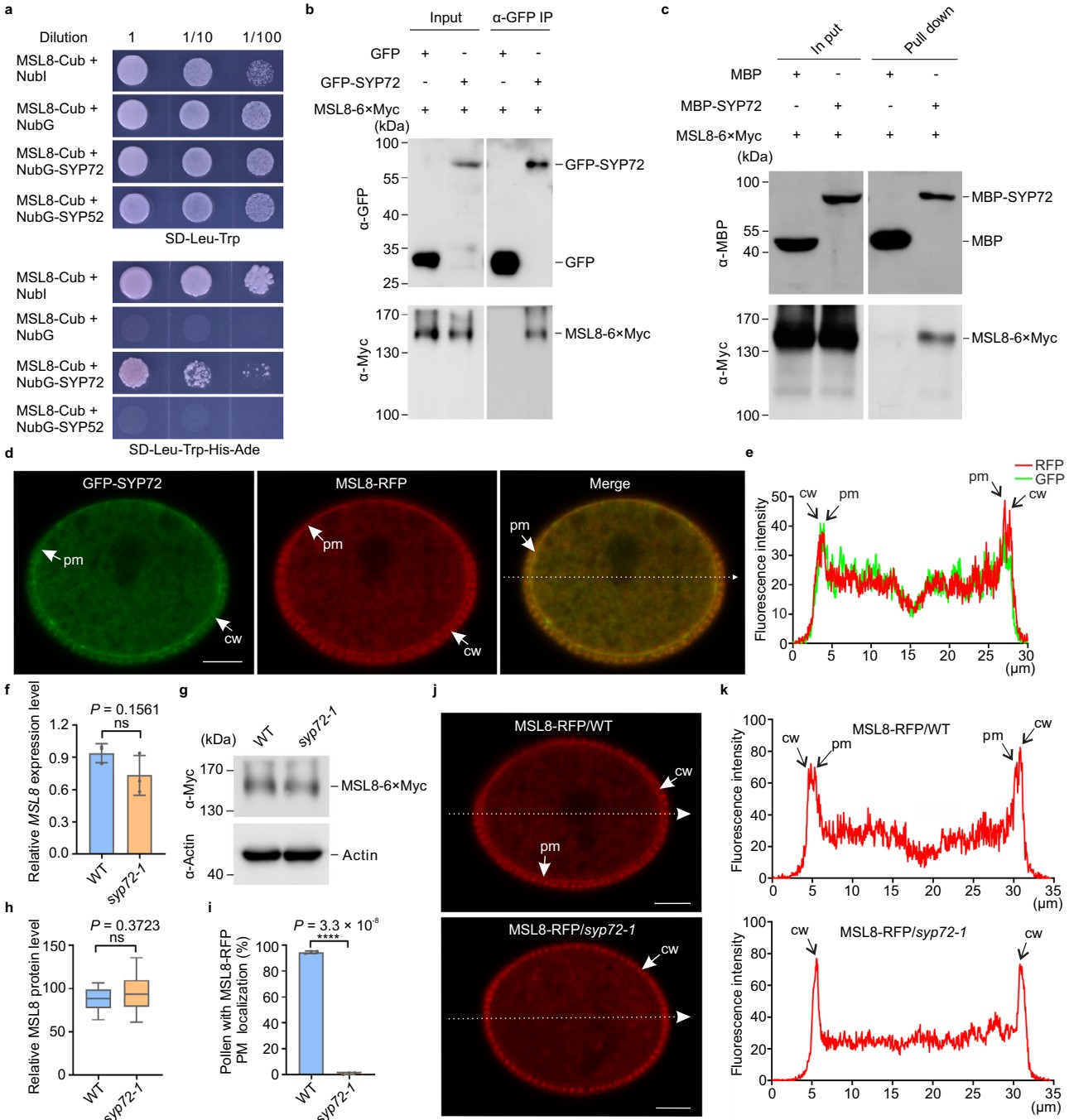

**Fig. 4 SYP72 directly binds to the mechanosensitive channel MSL8 and is required for its plasma membrane localization. a** Dual-membrane yeast two-hybrid assays with SYP72 and MSL8. **b** Co-immunoprecipitation (Co-IP) assays of GFP-tagged SYP72 and Myc-tagged MSL8. Arabidopsis flower buds expressing *GFP-SYP72/MSL8-6×Myc* or *GFP/MSL8-6×Myc* in pollen grains were used for Co-IP assay. **c** Pull-down assays between MBP-tagged SYP72 and Myc-tagged MSL8. **d**, **e** SYP72 strongly colocalizes with MSL8 in pollen. **d** Representative intracellular localization images of GFP-SYP72 and MSL8-RFP in pollen (*n* = 15 biologically independent pollen grains). Scale bars, 5 μm. **e** Profile of relative fluorescence signal intensity for GFP-SYP72 (green line) and MSL8-RFP (red line) along the dashed line indicated in **d**. **f–h** *MSL8* mRNA levels (**f**) and MSL8 protein levels (**g**, **h**) are not altered in the *syp72* mutant compared to those of WT plants. **f** RT-qPCR was performed to determine relative mRNA levels of *MSL8* in WT and *syp72-1*. mRNA levels of *MSL8* in the WT were set to 1. Data represent the mean ± SD (*n* = 3). **g** Western blot was used to determine protein levels of MSL8 in WT and *syp72-1*. Myc antibody was used to probe MSL8-6×Myc in WT and *syp72*-1 plants. Plant actin antibody was used as the control. This experiment was repeated 3 times with similar results. **h** Relative MSL8-RFP protein level was quantified according to the average RFP fluorescence intensity of pollen grains, with the level from WT plants set to 100. (*n* = 12 biologically independent pollen grains for WT and 20 for *syp72-1*). Box plots indicate the 25th, median, and 75th percentiles. The whiskers indicate the minimum and maximum values. **i** The percentages of pollen grains showing MSL8-RFP plasma membrane signals in WT and *syp72-1* plants. Data represent the mean ± SD from three independent experiments (*n* = 180 pollen grains). **j** The *syp72* mutation mislocalizes MSL8 in pollen (*n* = 16 biologically independent pollen grains). Scale bars, 5 μm. **k** Profile of relative fluorescence signal intensity for MSL8-RFP (red line) along the dashed lines (indicated in **j**) drawn across pollen from WT and *syp72-1* plants. cw cell wall, pm plasma membrane. (two-tailed Student's *t* test; ns, no significant difference, *P* > 0.05; ****P* < 0.0001). Yeast two-hybrid, Co-IP, pull-down, and RT-qPCR experiments were performed independently three times with similar results.

SYP73 proteins formed a relatively independent clade and were present only in the Brassicaceae (Supplementary Fig. 8b). SYP71 appeared to be the common ancestor of the SYP7 subfamily, as it was first identified in green algae and early land plants. As expected, we also identified conserved SNARE and transmembrane domains in ancestral SYP7 subfamily SNAREs from green algae and liverworts (Supplementary Fig. 8a).

Because only Brassicaceae plants have orthologues of Arabidopsis SYP72, we investigated whether these putative *SPY72* orthologues have similar molecular functions to Arabidopsis *SYP72*. To this end, we attempted interspecies genetic complementation experiments with *SYP72* genes from *B. napus* and *C. rubella* expressed in the Arabidopsis *syp72-1* mutant under the control of the Arabidopsis *SYP72* promoter. Both *BnSYP72* and *CrSYP72* successfully rescued the Arabidopsis *syp72-1* mutant pollen burst phenotype during rehydration and facilitated the relocalization of MSL8 to the plasma membrane, indicating that *BnSYP72* and *CrSYP72* are functional orthologues to Arabidopsis *SYP72* (Fig. 7b–i and Supplementary Fig. 10a–c).

Although *SYP72* counterparts have not been identified in plant lineages other than Brassicaceae, the majority of the green lineage, including green algae and early land plants, possess putative orthologues to Arabidopsis *SYP71*, a paralog of Arabidopsis *SYP72*, suggesting that *SYP71* is evolutionarily the ancestral form of the SYP7 subfamily. This raises the possibility that these SYP71 SNAREs from other green species might have functions similar to that of Arabidopsis SYP72 in assisting MSL8 localization. We therefore addressed the question that how conserved the ability to traffic MSL8 is in the SYP7 subfamily by expressing *SYP71* genes from various representative plant species under the control of the Arabidopsis *SYP72* promoter in the Arabidopsis *syp72-1* mutant. The evolutionarily most ancient *SYP71* gene, from the unicellular green alga *Chlamydomonas reinhardtii*, rescued the pollen bursting phenotype of Arabidopsis *syp72-1* mutant plants to WT levels, indicating full functionality. As expected, canonical *SYP71* members from the early land plant *M. polymorpha*, the moss *P. patens*, the lycophyte *Selaginella moellendorffi*, the gymnosperm Sitka spruce (*Picea sitchensis*), and several representative angiosperms, including the eudicot tobacco (*Nicotiana tabacum*) and the monocot rice (*Oryza sativa*), all similarly complemented the *syp72* pollen bursting phenotype and promoted MSL8 relocation to the plasma membrane (Fig. 7b–i and Supplementary Fig. 10). These successful interspecies complementation experiments suggest that SYP71 first appeared in unicellular algae, and then later, in the time since the separation of the Brassicaceae within the angiosperm lineage, its orthologue SYP72 evolved the specific function of regulating the localization of MSL8 to the plasma membrane, a critical step in pollen survival in the face of the hypoosmotic shock generated during rehydration.

## Discussion

The male gametophyte has evolved various mechanisms to ensure that pollen grains and pollen tubes maintain their cellular integrity during their long journey to transmit sperm cells to the female gamete. Genes expressed in the pollen tube and encoding components of a receptor-like kinase complex, including ANXUR1 (ANX1) and ANX2, BUDDHAS PAPER SEAL 1 (BUPS1) and BUPS2, LORELEI-LIKE-GPI ANCHORED PROTEIN 2 (LLG2) and LLG3, and cell wall leucine-rich repeat extensin (LRX) together with rapid alkalinization factor 4 (RALF4) and RALF19, have been reported to maintain pollen tube integrity within female tissues[15–18]. Whereas the process of pollen tube growth in female tissues is relatively well understood, it is largely unknown how desiccated pollen grains maintain their

cellular integrity upon hypoosmotic shock during their rehydration on the stigma. One pioneering study reported that the pollen-specific mechanosensitive channel MSL8 is required for pollen survival after this hypoosmotic shock. Compared to WT pollen grains, *msl8* pollen grains burst and lose cellular integrity during rehydration[6]. Here, we investigated the previously uncharacterized Qc-SNARE member SYP72, which helps position the mechanosensitive channel MSL8 at the plasma membrane, and obtained new knowledge of the role of Qc-SNAREs in helping maintain pollen cellular integrity in the face of hypoosmotic challenge during pollen rehydration. Like *MSL8*, *SYP72* was exclusively expressed in the male gametophyte and showed maximum expression in mature pollen, suggesting that SYP72 together with MSL8 form a pollen-specific pathway that enhances pollen survival of the osmotic forces released during rehydration.

Previous studies indicated that SNAREs are key elements driving membrane fusion and play critical roles in several fundamental processes such as cytokinesis[8,10,19] and maintenance of endoplasmic reticulum (ER) integrity[20]. Generally, SNAREs mediate membrane fusion between intracellular compartments and the plasma membrane, or between different intracellular compartments, by forming a complex between vesicle- and target-membrane-associated SNAREs[7]. This study demonstrates that, unlike most other Qc-SNARE proteins, SYP72 has a critical role in regulating the intracellular location of the mechanosensitive channel MSL8 through the formation of a SNARE–ion channel complex. MSL8 is a pollen-specific membrane-tension-gated ion channel that is required for pollen survival following the hypoosmotic shock of rehydration[6]. Our results revealed that MSL8 did not localize to the plasma membrane in loss-of-function *syp72* mutants upon rehydration, but LLG3, ANX1/2, and BUPS1 could still localize to the plasma membrane (Fig. 4i–k and Supplementary Fig. 11), implying that the SYP72-MSL8 pathway is required for pollen survival at this critical stage. Besides *msl8* and *syp72* mutants having similar phenotypes in pollen rehydration, *msl8* and *syp72* mutants also have different phenotypes, such as enhanced in vitro germination and transmission ratio of mutant male allele, suggesting that except the function in pollen rehydration, SYP72 and MSL8 might have some other different roles. Since the bursting rate of *msl8* pollen grains is lower than that of *syp72* mutant and the phenotype of *syp72 msl8* double mutant pollen resembles the *syp72* single mutant (Supplementary Fig. 12), there might be some other cargos such as lipids and other membrane proteins also contributing to the bursting phenotype of *syp72* pollen during rehydration. Notably, although SNARE proteins usually assemble into functional SNARE complexes that contain one copy each of Qa-, Qb-, Qc-, and R-SNARE proteins[8], the interaction partners of SYP72 among SNARE proteins in Arabidopsis are still unknown. Hence, elucidating the role of the SNARE–MSL8 complex in regulating pollen rehydration will require the identification of the SNARE complex components that associate with SYP72. Identification of other cargos of SYP72-containing SNARE complexes will also facilitate further understanding of the mechanism of SYP72 in pollen rehydration.

SNAREs are an evolutionarily conserved family of proteins required for membrane fusion in eukaryotic cells. However, the number of SNARE-encoding genes has increased markedly in the genomes of land plants as a result of both the expansion of members in conserved SNARE subfamilies and the appearance of plant-specific SNARE subfamilies, such as NPSN Qb-SNAREs and SYP7 Qc-SNAREs[7]. In Arabidopsis, *SYP72*, together with *SYP71* and *SYP73*, forms the plant-specific SYP7 SNARE subfamily, which evolved early in plant evolution[20]. *SYP71* orthologues have been identified in green algae, mosses, ferns, and gymnosperms (Supplementary Fig. 8b), whereas *SYP72* and

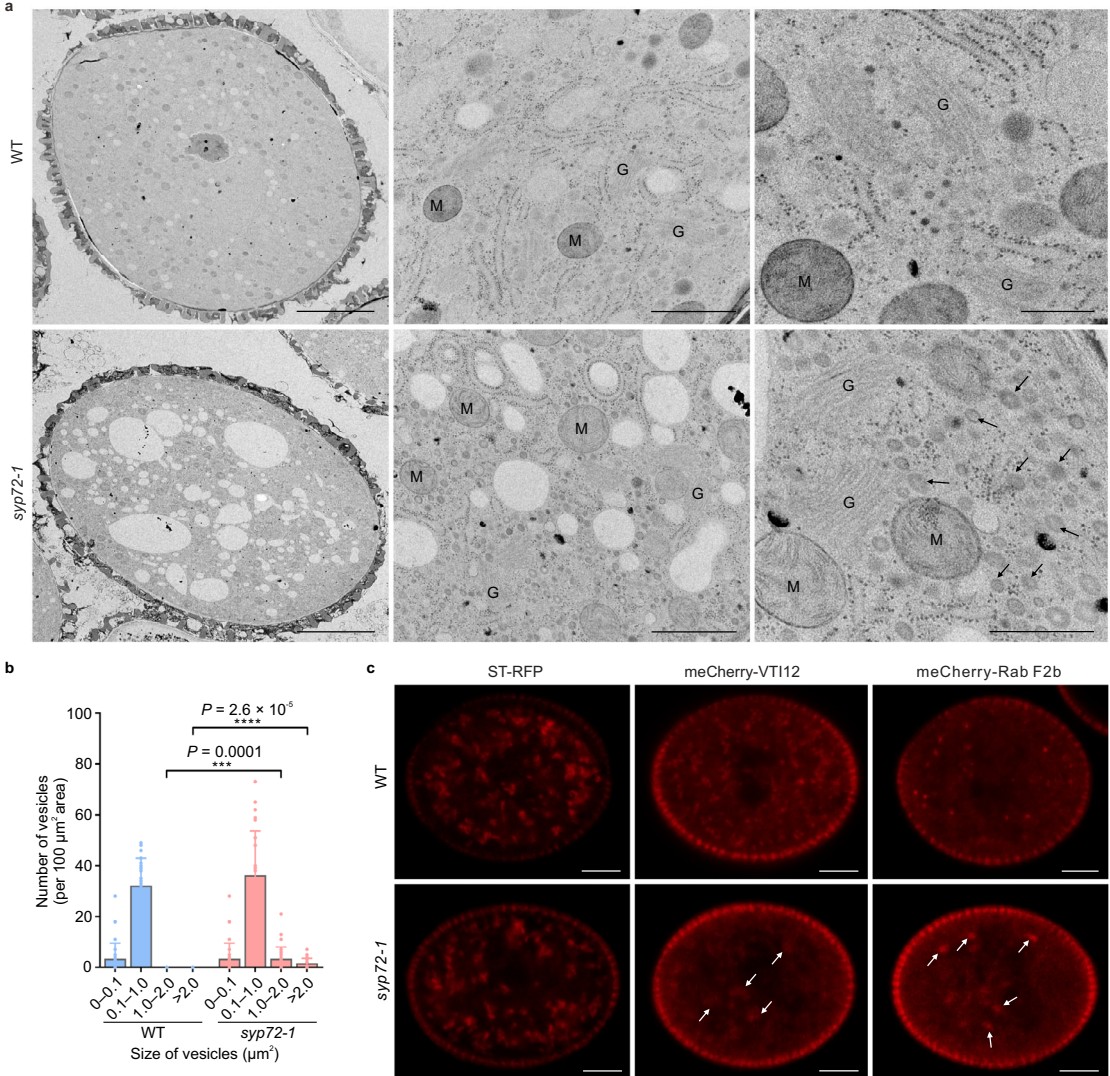

**Fig. 5 syp72 mutation leads to abnormal distribution of TGN/EE and endosomes in pollen. a** TEM analysis reveals the accumulation of endosome-like structures in *syp72-1* pollen (bars: left, 5 μm; middle, 1 μm; right, 500 nm; *n* = 30 biologically independent pollen grains). M mitochondrion, G Golgi apparatus. Arrows indicate numerous endosome-like structures in the *syp72-1* mutant. **b** The number of vesicles of different sizes in WT and *syp72-1* pollen. Data represent the mean ± SD from 30 pollen grains (*n* = 30). Two-tailed Student's *t* test was used for statistical analysis (***$P < 0.001$; ****$P < 0.0001$). **c** Representative images of pollen from WT and *syp72-1* plants expressing *ST-RFP* (Golgi apparatus; *n* = 10 biologically independent pollen grains), *mCherry-VTI12* (TGN/EE; *n* = 15 biologically independent pollen grains), and *mCherry-Rab F2b* (endosome; *n* = 13 biologically independent pollen grains). Arrows indicate abnormal accumulation of VTI12-labelled TGN/EE and Rab F2b-labelled late endosomes. Scale bars, 5 μm.

*SYP73* appear to be confined to the Brassicaceae (as well as *C. reinhardtii*). Considering the phylogenetic relationships among SYP7 family SNAREs from green algae to angiosperms, we propose that SYP71 is an ancient SNARE that was later duplicated to give rise to SYP72 and SYP73 during plant evolution.

Although *SYP71* and *SYP73* have specific roles in early root development, seed development, and virus infection, relative to *SYP72*, under normal conditions[20,21], ectopic expression of *SYP71* and *SYP73* in pollen rescued the male-sterile phenotype of *syp72* mutants, suggesting that SNARE proteins in the SYP7 subfamily have a conserved molecular function in regulating the intracellular location of the mechanosensitive channel MSL8, a theory that is also supported by their high sequence similarity and shared conserved motifs (Supplementary Fig. 8a). Moreover, similar to SYP72, SYP71 and SYP73 are also localized to the plasma membrane when ectopically expressed in pollen (Fig. 6b). This result raises the possibility that the functional diversity of SNARE proteins within the SYP7 subfamily may have resulted

from their possession of distinct promoter elements that ensure their differential spatio-temporal expression in plant development (Fig. 1a, b and Supplementary Fig. 2). In this scenario, even though the *SYP71* ancestor was already present in unicellular green algae, only plants belonging to the Brassicaceae will have gained a pollen-specific *SYP72* to ensure pollen survival after hypoosmotic shock through evolutionary changes to their *cis*-regulatory regions driving its pollen-specific expression. The distinct expression pattern and cis-regulatory sequences of Arabidopsis *SYP71*, *SYP72*, and *SYP73* support this hypothesis (Fig. 1a, b and Supplementary Figs. 2 and 13). In the future, it will be interesting to compare the expression patterns of *SYP7* subfamily genes in a given plant species and across different species to directly test this hypothesis.

## Methods

**Materials**. T-DNA insertion lines including *SALK_005288*, *SALK_132295C*, and *SALK_045561* were obtained from ABRC. T-DNA insertion mutant *msl8* (GK-

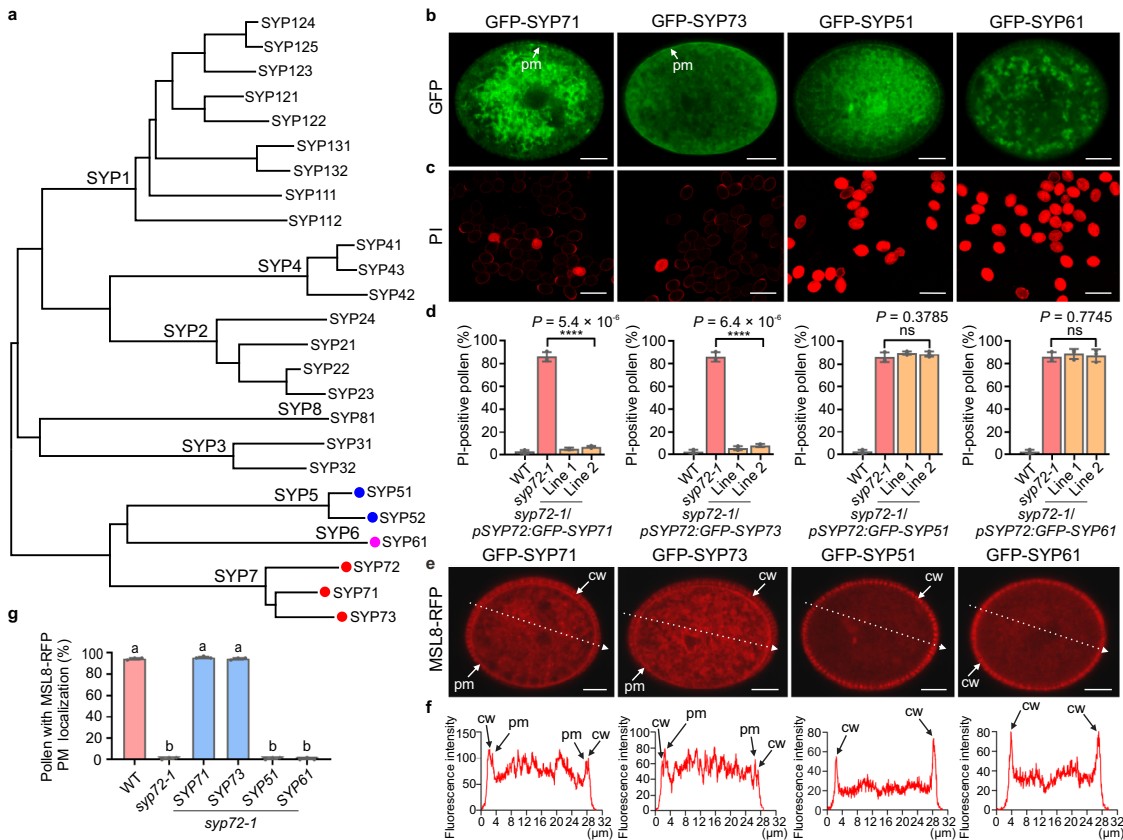

**Fig. 6 Arabidopsis SYP7 SNAREs have a conserved role in protecting rehydrating pollen from hypoosmotic shock. a** Phylogenetic relationship of Arabidopsis syntaxins. The tree was generated using the neighbour-joining method in MEGA7. The three members of the SYP7 subgroup are indicated with red dots. **b** Ectopic expression of *SYP7*, *SYP6*, and *SYP5* genes under the control of the *SYP72* promoter in *syp72-1* pollen (*n* = 10 biologically independent pollen grains). **c** Analysis of plasma membrane integrity during rehydration of pollen grains from the *syp72-1* mutant stably expressing Arabidopsis syntaxin genes *SYP71*, *SYP73*, *SYP51*, or *SYP61* under the control of the *SYP72* promoter, as observed by PI staining. Scale bars, 50 μm. **d** Percentages of PI-positive pollen grains from *syp72-1* transformed with Arabidopsis syntaxin genes *SYP71*, *SYP73*, *SYP51*, or *SYP61*. Two independent transgenic lines for each syntaxin gene were used for analysis. Data are the mean ± SD from three independent replicates. PI-positive pollen grains were scored from 300 pollen grains for each replicate. Two-tailed Student's *t* test was used for statistical analysis (ns no significant difference, *P* > 0.05; ****P* < 0.0001). **e** Subcellular localization of MSL8-RFP in the *syp72-1* mutant stably expressing different Arabidopsis syntaxin genes. Scale bars, 5 μm. **f** Profile of relative fluorescence intensity for MSL8-RFP (red line) along a dashed line (labelled in **e**) drawn across a pollen grain. **g** The percentages of pollen grains showing MSL8-RFP plasma membrane signals (**e**) in *syp72-1* plants transformed with different Arabidopsis syntaxin genes. Data represent the mean ± SD from three independent experiments, with 60 pollen grains analysed in each experiment (*n* = 180). Ordinary one-way ANOVA with Tukey's multiple comparisons test was used for statistical difference analysis between different groups. *P* < 0.05 was considered statistically significant. Different letters above the bars indicate significant differences between groups. cw cell wall, pm plasma membrane.

579H09) was obtained from the Nottingham Arabidopsis Stock Centre (NASC). Arabidopsis Columbia-0 (Col-0), T-DNA insertion lines, reporter lines including UBQ10, NPSN12-mCherry; UBQ10, mCherry-Rab A1e; UBQ10, mCherryRab A1g; UBQ10, mCherry-Rab F2a; UBQ10, mCherry-VAMP711; UBQ10, mCherry-Rab A5d; UBQ10, mCherry-Rab C1; UBQ10, mCherry-Rab D1; and UBQ10, mCherry-VTI12[12], and *Nicotiana benthamiana* were cultured in a greenhouse at 22 ± 1 °C, 16/8 h (light/dark).

**Sequence alignment and phylogenetic analysis**. Multiple sequence alignment of Arabidopsis SNARE protein sequences was performed with the ClustalW program, and a phylogenetic tree was constructed with MEGA 7.

**Vector construction and plant transformation**. To generate a gDNA complementation construct, a 3468-bp genomic fragment containing a 1236-bp promoter region, the transcribed region, and a 372-bp fragment downstream stop codon was amplified from the genomic DNA and cloned into pCAMBIA1300 carrying *LAT52:RFP*. To generate the SYP72 GFP fusion construct, *pSYP72:GFP-SYP72* was generated in the destination vector K32. Similarly, *pSYP72:GFP-SYP71*, *pSYP72:GFP-SYP73*, *pSYP72:GFP-SYP51*, *pSYP72:GFP-SYP61*, and *pSYP72::GFP-SYP7* (*SYP7* genes from different species) were also generated in the vector K32. *SYP71* or *SYP72* coding regions of *O. sativa*, *N. tabacum*, and *B. napus* were amplified from their respective cDNA. *SYP71* or *SYP72* coding regions of *C. reinhardtii*, *M. polymorpha*, *P. patens*, *S. moellendorffii*, *P. sitchensis*, and *C. rubella*

were synthesized (GenScript or Sangon Biotech). The *gMSL8-RFP* construct was generated according to a previous report by exchanging the GFP with RFP[6]. Primers for vector construction are listed in Supplementary Table 2. The DNA cassette for trans-Golgi marker ST-RFP was first generated according to a previous report[22], and then the whole cassette was subcloned into the destination vector K32 downstream of the promoter *RPL18aB*[23]. *pANX1:ANX1-GFP*, *pANX2:ANX2-GFP*, and *pBUPS1:BUPS1-GFP* were constructed according to the previous report[15,18]. The primers used for the vector construction are listed in Supplementary Table 2. All constructs were transformed into Arabidopsis using *Agrobacterium tumefaciens* strain GV3101 according to a previously published protocol[24].

**In vitro pollen germination analysis**. In vitro pollen germination assays were performed according to a previous protocol[25]. Pollen grains were cultured at 22 ± 1 °C in the germination medium which contains 5 mM CaCl₂, 1 mM Ca(NO₃)₂, 1 mM MgSO₄, 0.01% H₃BO₃, 18% sucrose, pH 7.0. After different hours of culture, pollen germination was observed under a microscope (Olympus IX71, Japan).

**RNA extraction and RT-qPCR**. Total RNA was extracted from the inflorescence tissue with the MiniBEST Plant RNA Extraction Kit (TaKaRa, China). cDNA was synthesized using an M-MLV First-Strand Kit (Invitrogen, USA) according to the manufacturer's instructions. RT-qPCR was performed in a 10 μl reaction mixture containing 5 μl 2×FastStart Essential DNA Green Master Mix (Roche, Germany),

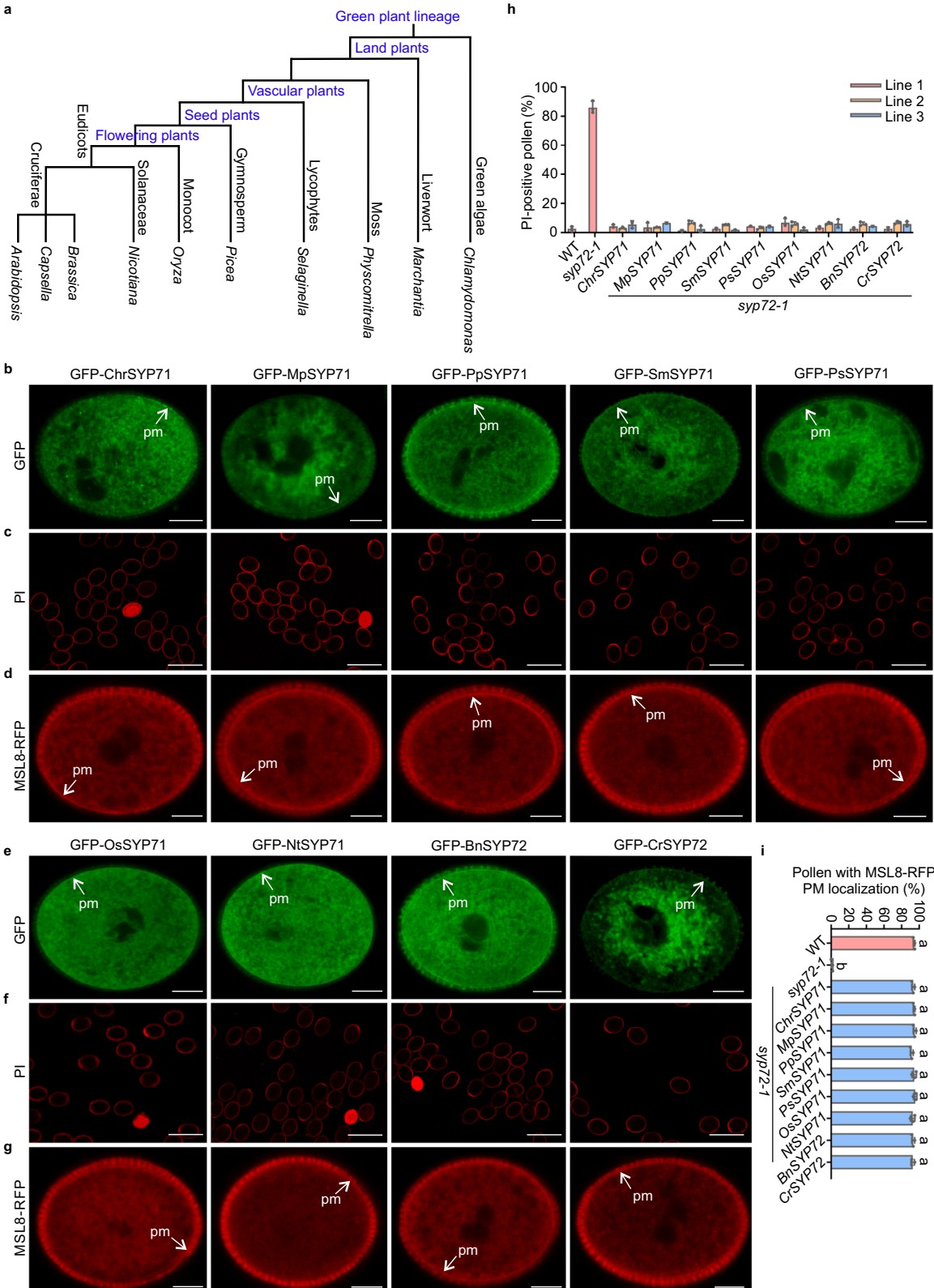

2 µl cDNA and 250 nM of each primer. The procedure for RT-qPCR was as follows: 95 °C for 10 min, and 40 cycles (95 °C for 15 s, annealing at 60 °C for 20 s, extension at 72 °C for 30 s) on a CFX Connect™ Real-Time System (Bio-Rad, USA). *AT1G58050*, *AT1G13320*, and *AT4G34270* were selected as reference genes. RT-qPCR data were analysed using the Bio-Rad CFX Manager v.3.1. Primers for RT-qPCR are listed in Supplementary Table 2.

**Yeast two-hybrid assay**. A yeast two-hybrid membrane protein system was used to verify the interaction between SYP72 and MSL8. The coding regions of *SYP52* and *SYP72* were cloned into the pPR3-N, whereas the coding regions of *MSL8* and *MSL9* were cloned into the pBT3-C. Yeast transformation and growth assays were performed according to the manufacturer's instructions (Dualsystem Biotech). Primer sequences for vector construction are listed in Supplementary Table 2.

**Fig. 7 Plant-specific SYP7 syntaxins have an evolutionarily conserved role in the survival of hypoosmotic shock during pollen rehydration. a** Schematic summary of major species of the green lineage, from green algae to angiosperms, is discussed here. **b, e** Expression of *SYP7* family members from green algae to angiosperms in the Arabidopsis *syp72-1* mutant (*n* = 10 biologically independent pollen grains). Scale bars, 5 μm. **c, f** Plasma membrane integrity of pollen grains from the *syp72-1* mutant stably transformed with *SYP7* family genes from major species of the green lineage listed in **a**, during rehydration, as revealed by PI staining. Scale bars, 50 μm. **d, g** Intracellular location of MSL8-RFP in the *syp72-1* mutant stably expressing *SYP7* family genes from different species. Scale bars, 5 μm. **h** Percentage of PI-positive pollen grains in different complementation lines. Data are the mean ± SD from three independent replicates. The percentage of PI-positive pollen grains was determined by scoring 300 pollen grains per replicate. **i** The percentages of pollen grains showing MSL8-RFP plasma membrane signals (**d, g**) in different complementation plants. Data represent the mean ± SD from three independent experiments, with 60 pollen grains analysed in each experiment (*n* = 180). Ordinary one-way ANOVA with Tukey's multiple comparisons test was used for statistical difference analysis between different groups. *P* < 0.05 was considered statistically significant. Different letters above the bars indicate significant differences between groups. *Chr Chlamydomonas reinhardtii, Mp Marchantia polymorpha, Pp Physcomitrium (Physcomitrella) patens, Sm Selaginella moellendorffii, Ps Picea sitchensis, Os Oryza sativa, Nt Nicotiana tabacum, Bn Brassica napus, Cr Capsella rubella,* pm plasma membrane.

**Co-IP, pull-down, and western blot assays.** The constructs *35S:GFP-SYP72* and *35S:MSL8-6×Myc* were transiently coexpressed in the leaves of *N. benthamiana* according to a previous protocol[26]. *35S:GFP* and *35S:MSL8-6×Myc* were also coexpressed in the *N. benthamiana* leaves as a negative control. Total proteins extracted from infiltrated leaves were used for subsequent Co-IP assay. To further confirm the interaction of SYP72 and MSL8 in vivo, transgenic plants expressing *pSYP72:GFP-SYP72, LAT52:GFP,* and *LAT52: MSL8-6×Myc* were generated respectively. *pSYP72:GFP-SYP72* and *LAT52:GFP* transgenic plants were crossed with *LAT52:MSL8-6×Myc* transgenic plants to generate *pSYP72:GFP-SYP72 LAT52:MSL8-6×Myc* and *LAT52:GFP LAT52: MSL8-6×Myc* plants. Flower buds containing mature pollen were collected for protein extraction and Co-IP assay. Co-IP was performed according to a previous report[27]. Infiltrated tobacco leaves or flowering buds were collected and ground into fine powder with liquid nitrogen. The powder was then resuspended in a buffer which contains 0.1 M Tris-HCl (pH 7.4), 150 mM NaCl, 1 mM EDTA, 10% glycerol, 0.5% NP-40, 0.5 mM dithiothreitol, 1 mM phenylmethylsulfonyl fluoride, and proteinase inhibitor cocktail (Roche). The extracts were centrifuged at 13,000 × *g* for 10 min at 4 °C, and the supernatants were collected for Co-IP assay. GFP-trap (Chromotek, Germany) was used to capture the GFP-tagged proteins according to the manufacturer's instructions. Detection of immunoprecipitates was performed as previously described using mouse anti-Myc (Abclonal, China; Catalogue number: AE010; 1:1,000 dilution) and mouse anti-GFP antibody (Abclonal, China; Catalogue number: AE012; 1:1,000 dilution), respectively. *MBP-SYP72* and *MBP* coding sequences were cloned into the vector Pet28a, and the resulting vectors were transformed into *Escherichia coli* BL21 (DE3) for recombinant protein expression. *MSL8-6×Myc* was expressed in the *N. benthamiana* leaves. Recombinant protein expression in *E. coli* BL21 (DE3) was induced with 0.2 mM isopropyl-β-d-thiogalactopyranoside for 18 h at 16 °C. Bacterial cells were then collected and lysed by sonication in a buffer that contains 20 mM Tris-HCl (pH 8.0), 150 mM NaCl, proteinase inhibitor cocktail (Roche). Total protein was extracted from infiltrated tobacco leaves according to the protocol for Co-IP mentioned above. Ni-NTA His Bind resin (Novagen, Germany) was then used for pull-down assay according to the manufacturer's instructions with minor modifications. The pull-down samples were subjected to sodium dodecyl sulfate polyacrylamide gel electrophoresis and then transferred to nitrocellulose membranes (Millipore) for immunoblotting with mouse anti-MBP (NEB; Catalogue number: E8032S; 1:1000 dilution) and mouse anti-Myc (Abclonal; Catalogue number: AE010; 1:1000 dilution).

To compare the protein levels of MSL8 in WT and *syp72*, *MSL8-6×Myc* transgenic plants were crossed with *syp72-1* to generate *syp72-1 MSL8-6×Myc* plants. Flower buds containing mature pollen were collected for protein extraction and western blot assay. Anti-bodies including mouse anti-Myc (Abclonal, China; Catalogue number: AE010, 1:1000 dilution) and mouse anti-Actin (Abbkine; Catalogue number: A01050; 1:1000 dilution) were used for western blotting assays.

**Aniline blue, PI-FDA, and Alexander's staining.** To investigate pollen germination in vivo, *syp72* and WT pollen grains were pollinated to the emasculated WT pistils, respectively. Siliques at different time points were harvested in Carnoy's fixative (ethanol:acetic acid = 3:1) for 12 h. Then, aniline blue staining was performed according to a protocol described previously[28]. After fixation, the pistils were washed with 50 mM phosphate-buffered saline (PBS) buffer three times. The pistils were then transferred into 1 M NaOH for softening. After extensive washing with PBS, the pistils were stained with 0.1% aniline blue in dark. The stained pistils were observed under a fluorescence microscope excited with the light of ultraviolet (Olympus IX71, Japan). To examine the viability of pollen grains, Alexander's staining was performed according to a previous protocol[29]. The anthers were fixed in the Carnoy's fixative for 2 hours, and then stained in Alexander's staining solution, which was prepared by adding the following components: 10 ml 95% alcohol, 1 ml malachite green (1% solution in 95% alcohol), 25 ml glycerol, 5 ml acid fuchsin (1% solution in water), 0.5 ml orange G (1% solution in water), 4 ml glacial acetic acid and 54.5 ml distilled water. After staining, the samples were observed under a microscope (Olympus IX71, Japan). To evaluate the plasma membrane integrity of pollen, we stained the pollen grains with 1 μg/ml PI (Sigma) and 2 μg/ml FDA (Sigma) or 1 μg/ml PI alone. After staining, pollens were observed under a confocal microscope using an HC PL APO CS2 ×20/0.75 dry

objective (Leica TCS SP8, Germany). PI was excited at 552-nm laser and the emission signals were collected at 600–660 nm. FDA was excited at 488-nm laser, and the emission was measured at 500–550 nm.

**Confocal laser scanning microscope (CLSM) observation and image analysis.** Pollen expressing different organelle markers were observed and photographed using a CLSM equipped with HyD detectors (Leica TCS SP8, Germany). Pollen grains were excited at 488-nm laser with emitted light measured at 498–550 nm for GFP observations and excited at 552-nm laser with emitted light measured at 594–650 nm for RFP or mCherry observations. The internal Leica HyD detectors were used to capture the emission signals. An HC PL APO CS2 100x/1.40 oil objective was used to observe mCherry- or RFP-labelled intracellular organelles in WT and *syp72* pollen grains. An HC PL APO CS2 ×63/1.40 oil or HC PL APO CS2 ×40/1.10 water objective was used for pollen observation in other experiments. The laser intensity was set as 1–10%, and the gain value was set as 10–200 for different GFP-tagged SNARE proteins in pollen grains. The laser intensity was set as 1–10%, and the gain value was set as 90–200 for different mCherry or RFP-tagged organelle markers. Images were obtained using the Leica Application Suite Interface (LAS X) software ver.3.5.5.19976. For PI-FDA staining and co-localization analysis, the sequential scanning mode of the LAS-X program is used to avoid bleeding through. To avoid the influences of the auto-fluorescence of pollen wall on co-localization analysis, the regions of pollen grains except cell wall were chosen for co-localization analysis. Pearson correlation coefficients were calculated using ImageJ 1.52a. The fluorescence intensity was measured using the LAS-X software ver.3.5.5.19976 (Leica, Germany).

**TEM analysis.** For TEM analysis, ultrathin sections were prepared according to our previous protocol[30] with minor modifications. Briefly, each sample was collected in a sample carrier with ultra-low gelling temperature agarose (Sigma-Aldrich, A2576) and frozen in a high-pressure freezer (Leica EM ICE). After freezing, the samples were freeze-substituted with a solution of 2% $OsO_4$ (SPI, USA) and 0.1% UA (Merk, Germany) dissolved in acetone using Leica EMAFS2 according to a previous freeze substitution protocol:[30] −90 °C 72 h, −90 °C to −60 °C 8 h, −60 °C 12 h, −60 °C to −30 °C 8 h, −30 °C 12 h, −30 °C to −0 °C 8 h, 0 °C 2 h. After substitution, samples were washed with fresh acetone three times and then embedded under a series of resin acetone mixtures (30–100% resin [SPI, 02680-AB]). Ultrathin sections (70 nm) were prepared using the Leica EM UC7 ultramicrotome and examined using a JEM-1400plus TEM (JEOL). Images were collected on a Gatan Rio 9 camera (Gatan Inc., USA) equipped with the Gatan Microscopy Suite (GMS) 3 acquisition software.

**Fragments per kilobase of transcript per million mapped reads (FPKM) calculation.** The expression levels of *Arabidopsis SNARE* genes in different vegetative and reproductive organs were determined using publicly-available RNA-seq datasets. Expression levels of each gene were quantified as the FPKM. Mapping reads to the Arabidopsis Col-0 genome and reference transcripts was performed using Bowtie 2. FPKM values calculation was performed using RSEM.

**Statistical analyses.** Bar, dot, and box–whisker plots were generated using GraphPad Prism v.8.4.2. Student's *t* test (two-sided), one-way analysis of variance with Tukey's multiple comparisons test, and Chi-square test were performed for statistical analysis using GraphPad Prism v.8.4.2 or IBM SPSS Statistics 23.

**Reporting summary.** Further information on research design is available in the Nature Research Reporting Summary linked to this article.

## Data availability

Publicly available RNA-seq data sets, including seedling (GSE32318), stem (GSE102694), root and rosette (GSE87760), carpel (GSE56326), floral bud (GSE45685), ovule (DRR044369), mature pollen (PRJNA194429), and pollen tube (GSE98145), were used for the expression-level analysis in this study. The seeds used in the experiments and any

other data related to the findings of this study are available from the corresponding authors upon request. Source data are provided with this paper.

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

## Acknowledgements
We thank Prof. Huixia Shou (Zhejiang University) for the DUAL membrane yeast system, Prof. Lijia Qu (Peking University) for mCherry-tagged marker lines, and Prof. Chao Li (East China Normal University) for the LLG3-GFP maker line. This work was supported by the National Natural Science Foundation of China (31991201, 32170346), the Fundamental Research Funds for the Central Universities (2042020kf0198), and the "Ten Thousand Talents Program for Young Talents".

## Author contributions
P.Z. and M.-x.Z. designed the experiments. X.Z., Y.Z., L.W., H.L., Y.G., and M.L. performed the experiments. X.Z. and P.Z. analysed the data. P.Z. and M.-x.S. wrote the manuscript. All authors contributed to the presentation and agreed on the manuscript before submission.

## Competing interests
The authors declare no competing interests.
