## [Peer Review File · Nature Communications]

SYP72 interacts with the mechanosensitive channel MSL8 to protect pollen from hypoosmotic shock during hydrationREVIEWER COMMENTS

Reviewer #1 (Remarks to the Author):

This is a nice paper with interesting conclusions. It has robust results, large numbers were used, it is very clearly written and well-attributed, and multiple alleles and complementation are employed for solid genetic conclusions. There is a nice correlation between SYT72 localization, MSL8 localization, and pollen grain integrity across the orthologs tested.

We have a few major concerns, but we believe they can be addressed without too much trouble:

1) The proposed defect in MSL8 plasma membrane localization in the *syt72* background in Figure 4 and elsewhere needs better support--one pollen grain is not enough. What percent of pollen grains with MSL8-RFP signal show PM localization in WT or *syt72* background? This level of analysis would also be nice for the other cases where MSL8-RFP localization is assayed.

2) The authors strongly imply that the pollen bursting phenotype of *syp72* mutants is caused by the lack of MSL8 at the plasma membrane. Because this is a key claim of the manuscript, the authors need to test this directly through genetics and/or explore alternative explanations.

- If the only role of *syp72* in pollen hydration is trafficking MSL8, then this can be tested by examining the bursting rate of *syp72 msl8* double mutant pollen, which should resemble that of the *syp72* single mutant. If the *syp72* mutant has effects in addition to its role in trafficking MSL8, then *msl8* and *syp72* will additively promote bursting after rehydration.

- Alternative explanations for an additive effect:

- o SYP72 may be responsible for the trafficking of another protein needed for hypoosmotic shock survival.

- o The delivery of lipids (by membrane fusion) by SYP72-containing SNARE complexes theoretically could be just as important for hypoosmotic shock survival than the delivery of cargo proteins. Extra membrane would decrease the chances of plasma membrane rupture during hypoosmotic shock. In growing pollen tubes hypoosmotic shock increases the rate of exocytosis (Zonia and Munnik 2008; J Exp Bot), and this could also happen during pollen rehydration .

- Outside of pollen rehydration, *msl8* and *syp72* mutants have different phenotypes, which the authors should acknowledge in their discussion:

- *msl8* pollen has enhanced in vitro germination (Hamilton et al 2015, Science) , compared to the reduced germination of *syp72*

- Transmission of the *msl8-4* mutant allele through the male germline is only slightly reduced (44% compared to 50% expected ratio), compared to a severe reduction in *syp72* alleles (2-7%).

These differences strongly suggests that *SYP72* has other effects in pollen than its effect on *MSL8* localization. One possibility might be that there is an accumulation of vesicles (likely TGN/EE) in *syp72* mutant pollen (lines 221-238).

3) The exact nature of the immunoprecipitation experiments is hard to find. The source of recombinant or tagged proteins should be clearly stated in the text or in the figure legends. Lines 197-198: Indicate that Co-IP was performed using *N. benthamiana* leaves transiently expressing GFP-SYP72 and *MSL8*-Myc.

Minor comments

Line 16: first step might rather be considered compatibility and hydration

In lines 201-202 the authors claim that the pull-down assays also confirmed the direct interaction of *MSL8* and *SYP72*. But because *MSL8*-RFP was purified from *N. benthamiana*, it is possible that other plant-specific proteins co-precipitated with it and mediated the interaction with bacterially-expressed *SYP72*.

Line 237-238, Line 358-359 these conclusions seem overstated.

However, the conclusion their claim that *MSL8* and *SYP72* form a complex (lines 79 and 368) in the discussion by mentioning the known ion channel-SNARE complex of *AKT1*, *KC1*, and *SYP121* (a Qa-SNARE) (Grefen et al 2010, Plant Cell). For this reason, they may consider removing references to this being “unexpected”.

Lines 45-47: Include the starting volume of the dessicated pollen grain, or put in terms of percent of volume change.

Figure S1 and Figure S2a show exactly the same data, but in a different format. Maybe it's ok since it's publicly available data, but it seems unnecessary.

It would be helpful in Figure 1a-b and Figure S2b-e to have images of non-transgenic controls to demonstrate how much of fluorescence in the GFP channel is autofluorescence of the pollen cell wall.

Address in figure legend or results why there is nuclear GFP-SYP72 fluorescence in Figure 1b.

Line 185: Because of the differences in in vivo phenotypes between *syp72* and *msl8*, change to “*syp72* mutant pollen exhibited a phenotype in distilled water similar to that of *msl8* mutants.”

Lines 201-202: Explain the difference between this pull-down assay and the above Co-IP.

Line 218: MSL8 does not have a cytoplasmic distribution; even in *syp72* background it likely remains in endosomes. Change to “uniform intracellular distribution.”

Lines 321-322: Change to: “We therefore addressed the question of how conserved the ability to traffic MSL8 is in the SYP7 subfamily by expressing...”

Line 357: Change “indicating” to “suggesting.” The authors have not yet demonstrated that SYP72 and MSL8 function in the same pathway.

Chi-squared analysis should be reported in Table S1.

Reviewer #2 (Remarks to the Author):

This manuscript (NCOMMS-21-15740-T) by Zhou et al. report the function of a SNARE component SYP72 in pollen hypoosmotic shock during hydration through direct mediating the plasma membrane (PM) trafficking of the mechanosensitive channel MSL8. The authors provide genetic and phenotypic evidence of *syp72* mutant, and checked the PM targeting of MSL8, and tested the direct interaction between SYP72 and MSL8. Finally, the authors performed genetic complementation assay with the orthologues of SYP72 from different plant species and found that they are functionally conserved. SNARE proteins are essential components in membrane fusion in eukaryotic cells, but the function of some components diverge with the gene duplication during evolution. Overall, this study identified a SNARE component SYP72 and found its role in pollen function. The data are solid and worth of publication. I have some concerns for the provided evidence and conclusions, which may be helpful for the manuscript improvement.

Major concerns:

1. In figure 1, no data on the transcript expression level of the three *syp72* alleles was shown, and the phenotypic analysis of *syp72*-2 and -3 is also missing in figure 2.
2. In figure 2, PI-FDA combined staining of the pollen showed the impaired pollen membrane integrity of the mature pollen, and relatively better membrane integrity of nondehiscent pollen of *syp72*. The membrane integrity elevated with the increase of PEG concentration. Based on this data, the authors draw the conclusion that the membrane burst at rehydration on germination in *syp72*. Here one drawback is that the *syp72*-1 pollen germinate and grow within the pistil as shown in supplemental figure 3, and it appears that the pollen tube growth rate is slower than the WT. *in vivo* pollen burst assay needs to be performed to validate the phenotype. This is important as at the *in vivo* condition, the stigma is not an environment as hypoosmotic as the pollen germination media *in vitro*. And the pollen tube growth phenotype should be mentioned in the main text in case to mislead the readers that SYP72 is specifically involved in pollen germination. And the aborted ovules were scattered among the viable ovules in figure 1f and figure S7, indicting other phenotypes of the *syp72* pollen tubes and other cargos are also affected in the vesicle trafficking. At this condition, other membrane cargos are also needed to be examined in the *syp72* to fully reveal the function of SYP72 and its specificity to different cargos.
3. In addition, the authors concluded that the *syp72* pollen burst upon germination, similar to *msl8*, while no pollen-bursting images or quantification under the microscopic bright field were shown in the figures. PI-FDA staining assay is indirect and not sufficient to conclude a burst phenotype.
4. In figure 4, in the interaction assay with yeast two-hybrid and protein pull-down/Co-IP, no negative control, like MSL10 and other SYPs, was used for the interaction specificity. This is usually necessary for membrane integral proteins, especially at the condition that these data were acquired in heterologous systems, not in pollen.

5. For TEM in figure 5a, the number and size of the larger vesicle bodies should be qualified in different slides and different area.

6. In figure 4g, the protein level of MSL8 in the *syp72* and WT was quantified with the RFP fluorescence. This is unacceptable. The different transgenic lines, different plants from the same line, different pollen from the same plants usually show different expression level of the exogenous protein. The fluorescent signal of different images make this assay implausible. Western blot of different transgenic lines is required to confirm that MSL8 protein is properly expressed in *syp72*.

Minor concerns:

Figure 1a was not mentioned in the main text.

Reviewer #3 (Remarks to the Author):

The manuscript focus on a interesting and novel aspect of pollen development and the initial steps of plant reproduction – the mechanisms regulating pollen grain hydration in the stigma. The authors perform an original functional characterization of SYP72 which includes genetic and phenotypical analysis, live cell imaging, Y2H, Transmission electron microscopy. The abstract is clear and summarizes the main findings. The introduction states the relevance of the theme and reference list seems comprehensive. Discussion and conclusions are appropriate although some aspects of results need to be clarified. Language is clear. Figures are adequately produced and fig legends “independent” of main text help their analysis. Supplementary info provided is relevant.

Although methods are generally well described, the settings used for confocal live imaging – which are central for this manuscript – are totally absent and this is my main criticism to this work because it compromises critical interpretation of data in figures 3-7. All results regarding protein localization (and most importantly co-localization) in cells that exhibit significant auto-fluorescence (such as pollen grains) require appropriate controls to confirm that signal detected is specific and not resulting from contamination and/or “bleed-through” between different channels. This is absent in this work so the reported overlapping (e.g. with Rab A1) and co-localization (with NPSN) cannot be established. The different intensity levels of exines in Figure 6b for example seems to suggest that automatic settings were used which would explain the significant differences in the graphs of Fig 6e (cell wall in GFP-SYP71 is almost double if compared to GFP-SYP61).

One interesting result that is reported but not analysed is the fact the SYP72 mutant pollen has only a 1.3% germination in vitro but in vivo, the authors report a 30% seed set. Considering that an osmotic issue is under study, this should be analysed further. Can germination be improved in vitro if osmoticum is changed? What was the germination % of pollen expressing other SYP7_ genes? Did pollen in vivo

germinated at higher % to account for 30% seed set? (it seems so from fig S3). Or was there also a problem in guidance/fertilization? (possible since MSL8 is a mechanosensitive membrane channel)

A minor aspect: the authors state that “SYP72 is responsible for trafficking MSL8”; I believe that “probably involved in...” would be more adequate.

Manuscript tracking number: NCOMMS-21-15740-T

Manuscript title: SYP72 interacts with the mechanosensitive channel MSL8 to protect pollen from hypoosmotic shock during hydration.

Response to Reviewer 1

Q1. This is a nice paper with interesting conclusions. It has robust results, large numbers were used, it is very clearly written and well-attributed, and multiple alleles and complementation are employed for solid genetic conclusions. There is a nice correlation between SYP72 localization, MSL8 localization, and pollen grain integrity across the orthologs tested.

Reply: Thanks for your appreciation of our manuscript. Your comments have greatly helped us to improve our manuscript.

Q2. The proposed defect in MSL8 plasma membrane localization in the *syp72* background in Figure 4 and elsewhere needs better support--one pollen grain is not enough. What percent of pollen grains with MSL8-RFP signal show PM localization in WT or *syp72* background? This level of analysis would also be nice for the other cases where MSL8-RFP localization is assayed.

Reply: The statistical data of pollen grains showing MSL8-RFP plasma membrane localization in WT and *syp72* background have been added in the revised Figure 4i. Statistical data for MSL8-RFP plasma membrane localization in Fig. 6e, and 7d, g have also been added in the revised Fig. 6g and 7i.

Q3. The authors strongly imply that the pollen bursting phenotype of *syp72* mutants is caused by the lack of MSL8 at the plasma membrane. Because this is a key claim of the manuscript, the authors need to test this directly through genetics and/or explore alternative explanations.

- If the only role of *syp72* in pollen hydration is trafficking MSL8, then this can be tested by examining the bursting rate of *syp72 ms18* double mutant pollen, which should resemble that of the *syp72* single mutant. If the *syp72* mutant has effects in addition to its role in trafficking MSL8, then *ms18* and *syp72* will additively promote bursting after rehydration.

- Alternative explanations for an additive effect:

- o SYP72 may be responsible for the trafficking of another protein needed for hypoosmotic shock survival.

- o The delivery of lipids (by membrane fusion) by SYP72-containing SNARE complexes theoretically could be just as important for hypoosmotic shock survival than the delivery of cargo proteins. Extra membrane would decrease the chances of plasma membrane rupture during hypoosmotic shock. In growing pollen tubes hypoosmotic shock increases the rate of exocytosis (Zonia and Munnik 2008; J Exp Bot), and this could also happen during pollen rehydration.

- Outside of pollen rehydration, *ms18* and *syp72* mutants have different phenotypes, which the authors should acknowledge in their discussion:

- *ms18* pollen has enhanced in vitro germination (Hamilton et al 2015, Science), compared to the reduced germination of *syp72*

- Transmission of the *msl8-4* mutant allele through the male germline is only slightly reduced (44% compared to 50% expected ratio), compared to a severe reduction in *syp72* alleles (2-7%).

These differences strongly suggests that SYP72 has other effects in pollen than its effect on MSL8 localization. One possibility might be that there is an accumulation of vesicles (likely TGN/EE) in *syp72* mutant pollen (lines 221-238).

Reply: Yes, we agree with you. Besides *msl8* and *syp72* mutants have similar phenotype in pollen rehydration, *msl8* and *syp72* mutants also have different phenotypes such as enhanced *in vitro* germination and transmission ratio of mutant male allele. To further investigate genetic relationship between SYP72 and MSL8, we generated *syp72 msl8* double mutant according to your suggestions. The results revealed that the bursting rate of *syp72 msl8* double mutant pollen is similar to that of the *syp72* single mutant, indicating that MSL8 is one of downstream cargos of SYP72-containing SNARE complexes (Supplementary Fig. 12). Since the bursting rate of *msl8* pollen is lower than that of *syp72* single mutant, there might be some other cargos such as lipids and other membrane proteins also contributing to the bursting phenotype of *syp72* pollen during rehydration. Relevant discussions about other possible targets of SYP72-containing SNARE complexes and potential function differences between SYP72 and MSL8 have also been added in the discussion section of revised manuscript.

Q4. The exact nature of the immunoprecipitation experiments is hard to find. The source of recombinant or tagged proteins should be clearly stated in the text or in the figure legends. Lines 197-198: Indicate that Co-IP was performed using *N. benthamiana* leaves transiently expressing GFP-SYP72 and MSL8-Myc.

Reply: The original Co-IP was performed using *N. benthamiana* leaves transiently expressing GFP-SYP72 and MSL8-6xMyc. The figure for this result has been moved into the supplementary material (Supplementary Fig. 6b). New Co-IP experiments using transgenic Arabidopsis plants expressing GFP-SYP72 and MSL8-6xMyc, which was generated from the cross between *pSYP72:GFP-SYP72* and *LAT52:MSL8-6xMyc* transgenic plants (Fig. 4b). The details for both immunoprecipitation experiments have been added in the methods and figure legends.

Q5. Line 16: first step might rather be considered compatibility and hydration

Reply: To meet the requirement of the length of abstract (no more than 150 words), we have removed this sentence in the revised manuscript.

Q6. In lines 201-202 the authors claim that the pull-down assays also confirmed the direct interaction of MSL8 and SYP72. But because MSL8-RFP was purified from *N. benthamiana*, it is possible that other plant-specific proteins co-precipitated with it and mediated the interaction with bacterially-expressed SYP72.

Reply: To further confirm the interaction, Co-IP experiments using transgenic Arabidopsis plants expressing GFP-SYP72 and MSL8-6xMyc have been performed according to the suggestion from reviewer 2. The new results further confirmed the interaction of MSL8

and SYP72 (Fig. 4b). To express more precise about the pull-down assays in the manuscript, the word “direct” has removed in the revised manuscript.

Q7. Line 237-238, Line 358-359 these conclusions seem overstated.

However, the conclusion their claim that MSL8 and SYP72 form a complex (lines 79 and 368) in the discussion by mentioning the known ion channel-SNARE complex of AKT1, KC1, and SYP121 (a Qa-SNARE) (Grefen et al 2010, Plant Cell). For this reason, they may consider removing references to this being “unexpected”.

Reply: The words “unexpected” in lines 32, 79 and 368 have been removed according to your suggestion.

Q8. Lines 45-47: Include the starting volume of the desiccated pollen grain, or put in terms of percent of volume change.

Reply: The original paper (Rozier, F. et al 2020) did not report the starting volume of the desiccated pollen grain. So, the volume change during hydration is expressed as the percent according to your suggestion in the revised manuscript.

Q9. Figure S1 and Figure S2a show exactly the same data, but in a different format. Maybe it's ok since it's publicly available data, but it seems unnecessary.

Reply: Figure S2a has been removed in the revised supplementary material according to your suggestion.

Q10. It would be helpful in Figure 1a-b and Figure S2b-e to have images of non-transgenic controls to demonstrate how much of fluorescence in the GFP channel is auto fluorescence of the pollen cell wall.

Reply: We have provided the images of non-transgenic controls for *pSYP72:H2B-GFP*, *pSYP72:GFP-SYP72*, *pSYP71:GFP-SYP71*, *pSYP73:GFP-SYP73* in the revised Supplementary Fig. 2. The nucleolus of microspore and bicellular pollen have weak auto fluorescence. The auto fluorescence of the pollen cell wall has also been indicated in Fig. 1 and Supplementary Fig. 2.

Q11. Address in figure legend or results why there is nuclear GFP-SYP72 fluorescence in Figure 1b.

Reply: There is weak auto fluorescence in the nucleolus of microspore and bicellular pollen as shown in the images of non-transgenic controls acquired at the same confocal imaging conditions (Supplementary Fig. 2). The nuclear GFP-SYP72 fluorescence in Fig. 1b is also the auto fluorescence of nucleolus, which has been indicated in the figure legends.

Q12. Line 185: Because of the differences in *in vivo* phenotypes between *syp72* and *msl8*, change to “*syp72* mutant pollen exhibited a phenotype in distilled water similar to that of *msl8* mutants.”

Reply: The sentence “The *syp72* mutant exhibited a phenotype similar to that of *msl8* mutants” has been changed as “The *syp72* mutant pollen exhibited a phenotype in

distilled water similar to that of *msl8* mutants.”

Q13. Lines 201-202: Explain the difference between this pull-down assay and the above Co-IP.

Reply: For pull down assay, *MBP-SYP72* was expressed in *Escherichia coli* BL21 (DE3), and *MSL8-6xMyc* was expressed in the tobacco leaves. Total proteins were then extracted from *E. coli* expressing *MBP-SYP72* and leaves expressing *MSL8-6xMyc* for pull-down assay. For Co-IP, *GFP-SYP72* and *MSL8-6xMyc* were co-expressed in the tobacco leaves. Total proteins extracted from leaves expressing both of the *GFP-SYP72* and *MSL8-6xMyc* were used for Co-IP analysis. In the revised manuscript, Co-IP experiments using transgenic Arabidopsis plants expressing *GFP-SYP72* and *MSL8-6xMyc* were performed again according to the suggestion from reviewer 2, further confirming the interaction between SYP72 and MSL8. The details for pull-down and Co-IP assays have been added in the figure legends and methods of revised manuscript.

Q14. Line 218: MSL8 does not have a cytoplasmic distribution; even in *syp72* background it likely remains in endosomes. Change to “uniform intracellular distribution.”

Reply: “uniform cytoplasmic distribution” has been revised as “uniform intracellular distribution”.

Q15. Lines 321-322: Change to: “We therefore addressed the question of how conserved the ability to traffic MSL8 is in the SYP7 subfamily by expressing...”

Reply: This sentence has been revised as “We therefore addressed the question of how conserved the ability to traffic MSL8 is in the SYP7 subfamily by expressing...”

Q16. Line 357: Change “indicating” to “suggesting.” The authors have not yet demonstrated that SYP72 and MSL8 function in the same pathway.

Reply: The word “indicating” has been revised as “suggesting”.

Q17. Chi-squared analysis should be reported in Table S1.

Reply: Chi-squared analysis results has been added in the Table S1.

Response to Reviewer 2

Q1. This manuscript (NCOMMS-21-15740-T) by Zhou et al. report the function of a SNARE component SYP72 in pollen hypoosmotic shock during hydration through direct mediating the plasma membrane (PM) trafficking of the mechanosensitive channel MSL8. The authors provide genetic and phenotypic evidence of *syp72* mutant, and checked the PM targeting of MSL8, and tested the direct interaction between SYP72 and MSL8. Finally, the authors performed genetic complementation assay with the orthologues of SYP72 from different plant species and found that they are functionally conserved. SNARE proteins are essential components in membrane fusion in eukaryotic cells, but the function of some components diverge with the gene duplication during evolution. Overall, this study identified a SNARE component SYP72 and found its role in pollen function. The data are solid and worth of publication. I have some concerns for the provided evidence and conclusions, which may be helpful for the manuscript improvement.

Reply: Thanks for your appreciation of our work. Your comments have greatly helped us to improve the manuscript.

Q2. In figure 1, no data on the transcript expression level of the three *syp72* alleles was shown, and the phenotype analysis of *syp72-2* and *-3* is also missing in figure 2.

Reply: RT-PCR has been performed to examine the transcript levels of SYP72 in three *syp72* mutants. The results have been added in the revised Supplementary Fig. 3b. PI-FDA staining has been performed to assess plasma membrane integrity of mature pollen grains from *syp72-2* and *-3* during hydration. Statistical data for ruptured pollen grains in hydration were shown in Fig. 2b.

Q3. In figure 2, PI-FDA combined staining of the pollen showed the impaired pollen membrane integrity of the mature pollen, and relatively better membrane integrity of nondehiscent pollen of *syp72*. The membrane integrity elevated with the increase of PEG concentration. Based on this data, the authors draw the conclusion that the membrane burst at rehydration on germination in *syp72*. Here one drawback is that the *syp72-1* pollen germinate and grow within the pistil as shown in supplemental figure 3, and it appears that the pollen tube growth rate is slower than the WT. *in vivo* pollen burst assay needs to be performed to validate the phenotype. This is important as at the *in vivo* condition, the stigma is not an environment as hypoosmotic as the pollen germination media *in vitro*. And the pollen tube growth phenotype should be mentioned in the main text in case to mislead the readers that SYP72 is specifically involved in pollen germination.

Reply: According to your suggestion, *in vivo* pollen burst assay has been performed to validate the phenotype of *syp72* mutants. The results demonstrated that *syp72* pollen also ruptured in the pistil during the process of hydration, further confirming the results. New data have been added in the revised Fig. 2d-f.

We have also analyzed pollen tube growth phenotype of *syp72* mutants under *in vivo* and *in vitro* conditions. The results indicate that *syp72* pollen tubes grow slower than the wild type pollen tubes. Relevant data have been added in the revised Fig. 1g and Supplementary Fig. 4h, and this phenotype of *syp72* has also been introduced in the revised manuscript.

Q4. And the aborted ovules were scattered among the viable ovules in figure 1f and figure S7, indicating other phenotypes of the *syp72* pollen tubes and other cargos are also affected in the vesicle trafficking. At this condition, other membrane cargos are also needed to be examined in the *syp72* to fully reveal the function of SYP72 and its specificity to different cargos.

Reply: Yes, you are right. MSL8 is one of membrane cargo of SYP72 in pollen hydration, there might be other cargos including lipids and other membrane proteins in pollen and pollen tube. According to your suggestion, we have also investigated the trafficking of several other membrane cargos including LLG3, ANX1/2 and BUPS1 in *syp72*. The results revealed that LLG3, ANX1/2 and BUPS1 could still localize at the plasma membrane (Supplementary Fig. 11). Hence, identifying other cargos of SYP72 could be another independent project and will take years of efforts, which is worthy to be investigated through comprehensive omics studies in the future studies. In the present study, we focused on the role of SYP72 in pollen hydration and its relationship to MSL8.

Q5. In addition, the authors concluded that the *syp72* pollen burst upon germination, similar to *msl8*, while no pollen-bursting images or quantification under the microscopic bright field were shown in the figures. PI-FDA staining assay is indirect and not sufficient to conclude a burst phenotype.

Reply: Pollen-bursting images of *syp72* pollen under the microscopic bright field during hydration have been provided in the revised Fig. 2c according to your suggestion.

Q6. In figure 4, in the interaction assay with yeast two-hybrid and protein pull-down/Co-IP, no negative control, like MSL10 and other SYPs, was used for the interaction specificity. This is usually necessary for membrane integral proteins, especially at the condition that these data were acquired in heterologous systems, not in pollen.

Reply: Negative controls including SYP51 and MSL9 have been used in the yeast two-hybrid assay according to your suggestion. The relevant results have been added in the revised Fig. 4a and Supplementary Fig. 6a.

To further confirm the interaction of SYP72 and MSL8 *in vivo*, Co-IP experiments using transgenic plants expressing GFP-SYP72 and MSL8-Myc. Transgenic plants expressing *pSYP72:GFP-SYP72*, *LAT52:GFP* (negative control) and *LAT52:MSL8-6xMyc* were generated respectively. *pSYP72:GFP-SYP72* and *LAT52:GFP* transgenic plants were crossed with *LAT52:MSL8-6xMyc* transgenic plant to generate *pSYP72:GFP-SYP72/LAT52:MSL8-6xMyc* and *LAT52:GFP/LAT52:MSL8-6xMyc* plants, respectively. The flower buds containing mature pollen were collected for Co-IP assay, further confirming the interaction between SYP72 and MSL8. The new results have been added in the Fig. 4b.

Q7. For TEM in figure 5a, the number and size of the larger vesicle bodies should be qualified in different slides and different area.

Reply: The number and size of the larger vesicle bodies have been qualified. New statistical data have been added in the revised Fig. 5b.

Q8. In figure 4g, the protein level of MSL8 in the *syp72* and WT was quantified with the RFP fluorescence. This is unacceptable. The different transgenic lines, different plants from the same line, different pollen from the same plants usually show different expression level of the exogenous protein. The fluorescent signal of different images make this assay implausible. Western blot of different transgenic lines is required to confirm that MSL8 protein is properly expressed in *syp72*.

Reply: To compare the level of MSL8 in WT and *syp72*, we crossed MSL8-RFP transgenic plants with *syp72* mutant. The RFP fluorescence of MSL8-RFP in same transgenic line from the WT and *syp72* background was quantified, respectively. Statistical data of fluorescence intensities show no significant differences between WT and *syp72*.

To further confirm this result, western blot analysis was performed according to your suggestion. *MSL8-6xMyc* transgenic plants were crossed with *syp72* to generate *syp72/MSL8-6xMyc* plants. Western blot analysis was used to compare the protein level of MSL8-6xMyc in WT and *syp72* mutants using Myc antibody. The new data have been added in the revised Fig. 4g, which shows no visible differences between WT and *syp72*.

Q9. Figure 1a was not mentioned in the main text.

Reply: Thanks for your careful reading. Fig. 1a has been cited in the revised manuscript.

Response to Reviewer 3

Q1. The manuscript focus on a interesting and novel aspect of pollen development and the initial steps of plant reproduction – the mechanisms regulating pollen grain hydration in the stigma. The authors perform an original functional characterization of SYP72 which includes genetic and phenotypical analysis, live cell imaging, Y2H, Transmission electron microscopy. The abstract is clear and summarizes the main findings. The introduction states the relevance of the theme and reference list seems comprehensive. Discussion and conclusions are appropriate although some aspects of results need to be clarified. Language is clear. Figures are adequately produced and fig legends “independent” of main text help their analysis. Supplementary info provided is relevant.

Reply: Thanks for your appreciation of our manuscript. Your comments have greatly helped us to improve the manuscript.

Q2. Although methods are generally well described, the settings used for confocal live imaging – which are central for this manuscript – are totally absent and this is my main criticism to this work because it compromises critical interpretation of data in figures 3-7.

Reply: The settings for confocal microscope imaging have been added in the methods section of revised manuscript according to your suggestion.

Q3. All results regarding protein localization (and most importantly co-localization) in cells that exhibit significant auto-fluorescence (such as pollen grains) require appropriate controls to confirm that signal detected is specific and not resulting from contamination and/or “bleed-through” between different channels. This is absent in this work so the reported overlapping (e.g. with Rab A1) and co-localization (with NPSN) cannot be established.

Reply: Yes, you are right. It is an important point for co-localization analysis in pollen due to the auto-fluorescence of pollen wall. To achieve co-localization analysis in pollen, we improved the methods from three aspects. First, for possible bleed-through, we selected the sequential scanning mode in the Leica Application Suite X software to acquire images. By setting up a sequential scan, only one of the lasers scans the sample at a time to avoid bleed-through between GFP and RFP channels; Second, for the auto-fluorescence of pollen wall, the images of the central section of the pollen grains were acquired during the experiments, which could efficiently avoid the influences of auto-fluorescence from the top and bottom pollen wall. And the regions of the images of pollen grains except cell wall were chosen for correlation coefficient analysis, which could further exclude the influences of periphery pollen wall on co-localization analysis. Third, several markers including mCherry-VAMP711 and ST-RFP were used as the control for co-localization analysis (Fig. 3e and Supplementary Fig. 5e). These technical improvements ensure the reliability of co-localization analysis results in the manuscript. Detailed procedures for co-localization analysis have been added in the methods section of revised manuscript.

Q4. The different intensity levels of exines in Figure 6b for example seems to suggest that automatic settings were used which would explain the significant differences in the graphs of Fig 6e (cell wall in GFP-SYP71 is almost double if compared to GFP-SYP61).

Reply: The four images in Figure 6b are representative pollen images of four different *SYP* genes transgenic plants. Since there are different *SYP* genes, their expression levels and subcellular distributions in pollen grains are different. It is difficult to use same settings for imaging pollen grains expressing different *SYP* genes. To obtain their representative images, images were acquired under optimal system settings for each *SYP* gene.

Fig. 6e aimed to display whether MSL8-RFP could localize to the plasma membrane in *syp72* mutants when transformed with different Arabidopsis *SYP* genes. The images were acquired for displaying RFP signals in the plasma membrane under optimal system settings. The intensity of auto-fluorescence of pollen wall is not a key factor for this experiment.

Q5. One interesting result that is reported but not analysed is the fact the *SYP72* mutant pollen has only a 1.3% germination *in vitro* but *in vivo*, the authors report a 30% seed set.

Reply: Thanks for your careful reading. There are two possible reasons for this result. First,

In Arabidopsis, each wild type silique contains 50-60 seeds, whereas each *syp72* silique usually contain about 15 seeds (about 30% seed setting). However, the number of pollen grains in each flower is much higher than the number required for fertilization. There are hundreds of pollen grains in each Arabidopsis anther (Yan et al. 2017; Xu et al. 2020). The higher number of pollen grains in the anthers is one possible reason for the 30% seed set in *syp72* mutant. Second, we only know the *in vitro* germination ratio (1.3%) of *syp72* pollen grains, but could not know the accurate *in vivo* germination ratio of *syp72* pollen grains since the exact number of pollen grains and pollen tubes are difficult to be quantified in the pistil. Maybe there are some differences in the germination ratio between *in vitro* and *in vivo* since *in vitro* germination ratio of wild type pollen grains could only reach to about 85%.

Q6. Considering that an osmotic issue is under study, this should be analysed further. Can germination be improved *in vitro* if osmoticum is changed?

Reply: That's a nice suggestion. We have tried to change the osmoticum of medium by adding sucrose or PEG 3350 to the medium. However, the germination of both wild type and *syp72* pollen grains were significantly reduced. When the concentration of sucrose increased to 30%, no pollen could germinate. This result is consistent with the previous report (Boavida and McCormick 2007). The components of medium are critical for *in vitro* Arabidopsis pollen germination. Increasing the osmoticum of culture medium seems unfavorable for pollen germination.

Q7. What was the germination % of pollen expressing other *SYP7_* genes?

Reply: *In vivo* pollen germination of *syp72* pollen grains expressing other *SYP7* genes has been determined through aniline blue staining. Consistent with increased seed setting, *syp72* pollen grains expressing *SYP71*, *SYP73* and *SYPs* from other species germinate well in the pistils, whereas germination of *syp72* pollen grains expressing *SYP51* and *SYP61* display no obvious differences with *syp72* mutants. The new results have been added in the revised Supplementary Fig.9a and 10a.

Q8. Did pollen *in vivo* germinated at higher % to account for 30% seed set? (it seems so from fig S3). Or was there also a problem in guidance/fertilization? (possible since MSL8 is a mechanosensitive membrane channel)

Reply: Thanks for your careful reading. First, we would like to explain the differences between these two experiments. Seed setting was determined using self-pollinated siliques, whereas hand pollinated pistils were used for Aniline blue staining assay. There may be some differences in the number pollen grains for the pollination.

Second, accurate pollen germination ratio *in vivo* is hard to figure out since the exact number of pollen grains and pollen tubes in the pistil are difficult to be quantified. We could not know the accurate germination ratio of WT and *syp72* in the pistil. Third, pollen tube guidance of *syp72* pollen tube was analyzed according to your suggestion. The results revealed no visible defects in pollen tube guidance. New data have been added in the revised Supplementary Fig. 4i.

Q9. A minor aspect: the authors state that “SYP72 is responsible for trafficking MSL8”; I believe that “probably involved in...” would be more adequate.

Reply: Thanks for your suggestion. “responsible for” has been revised as “involved in” in the revised manuscript.

References:

1. Xu, X. F., X. X. Qian, K. Q. Wang, Y. H. Yu, Y. Y. Guo, X. Zhao, B. Wang, N. Y. Yang, J. R. Huang, and Z. N. Yang. 2020. Slowing Development Facilitates Arabidopsis mgt Mutants to Accumulate Enough Magnesium for Pollen Formation and Fertility Restoration. *Front Plant Sci*, 11: 621338.
2. Yan, J., J. C. Chia, H. Sheng, H. I. Jung, T. O. Zavodna, L. Zhang, R. Huang, C. Jiao, E. J. Craft, Z. Fei, L. V. Kochian, and O. K. Vatamaniuk. 2017. Arabidopsis Pollen Fertility Requires the Transcription Factors CITF1 and SPL7 That Regulate Copper Delivery to Anthers and Jasmonic Acid Synthesis. *Plant Cell*, 29: 3012-29.
3. Boavida, L. C., and S. McCormick. 2007. Temperature as a determinant factor for increased and reproducible in vitro pollen germination in *Arabidopsis thaliana*. *Plant Journal*, 52: 570-82.
4. Rozier, F., L. Riglet, C. Koderer, V. Bayle, E. Durand, J. Schnabel, T. Gaude, and I. Fobis-Loisy. 2020. Live-cell imaging of early events following pollen perception in self-incompatible *Arabidopsis thaliana*. *J Exp Bot*, 71: 2513-26.

REVIEWERS' COMMENTS

Reviewer #1 (Remarks to the Author):

The authors have addressed my original concerns and I'm satisfied with this version of the manuscript. I thank them for taking such care and effort with their revision!

Reviewer #2 (Remarks to the Author):

This revised version of the manuscript (#NCOMMS-21-15740A) addressed all my concerns and is readily publishable.

Reviewer #3 (Remarks to the Author):

The authors have addressed my main criticism and they seemed to also satisfactorily answered the comments raised by the other reviewers. As such, the revised manuscript reflects a more comprehensive analysis of the topic.

Having said that, I still find that the methods description concerning the imaging experiments lacks important details. The sequential scanning mode of SP8 is effective eliminating bleed-through IF the detected wavelength range (emission) is narrowed. So the methods can easily be complemented with the emission wavelength ensuring the reader that appropriate methods were used.

Similarly, I think it's essential that authors refer to laser intensity and gain settings. The autofluorescence of the CW in the pollen grain should not vary much and yet, those changes are obvious in the images displayed. This is likely to result from images collected with different gain and laser intensity. Such procedure is not necessarily a problem (or cause of artefacts) as even in the best conditions, X-FP protein expression levels can vary. Therefore, it will be important for readers to know how broad were laser intensity and gain setting levels.

All these values (gain, laser, emission wavelength) are embedded in the original Leica files so it will be very easy for authors to provide them.

Response to Reviewers :

Response to Reviewer #1

Q1. The authors have addressed my original concerns and I'm satisfied with this version of the manuscript. I thank them for taking such care and effort with their revision!

Reply : Thanks for your professional review and agreeing with the revision.

Response to Reviewer #2

Q2. This revised version of the manuscript (#NCOMMS-21-15740A) addressed all my concerns and is readily publishable.

Reply : Thanks for your professional review and agreeing with the revision.

Response to Reviewer #3

Q1. The authors have addressed my main criticism and they seemed to also satisfactorily answered the comments raised by the other reviewers. As such, the revised manuscript reflects a more comprehensive analysis of the topic.

Having said that, I still find that the methods description concerning the imaging experiments lacks important details. The sequential scanning mode of SP8 is effective eliminating bleed-through IF the detected wavelength range (emission) is narrowed. So the methods can easily be complemented with the emission wavelength ensuring the reader that appropriate methods were used.

Similarly, I think it's essential that authors refer to laser intensity and gain settings. The autofluorescence of the CW in the pollen grain should not vary much and yet, those changes are obvious in the images displayed. This is likely to result from images collected with different gain and laser intensity. Such procedure is not necessarily a problem (or cause of artefacts) as even in the best conditions, X-FP protein expression levels can vary. Therefore, it will be important for readers to know how broad were laser intensity and gain setting levels.

All these values (gain, laser, emission wavelength) are embedded in the original Leica files so it will be very easy for authors to provide them.

Reply: Thanks for your comments again. The detailed settings including objective, laser intensity, gain value and emission wavelength for CLSM observation have been added in the method section of revised manuscript.